# Genomic Characterization of *Parengyodontium torokii* sp. nov., a Biofilm-Forming Fungus Isolated from Mars 2020 Assembly Facility

**DOI:** 10.3390/jof8010066

**Published:** 2022-01-09

**Authors:** Ceth W. Parker, Marcus de Melo Teixeira, Nitin K. Singh, Huzefa A. Raja, Kristof B. Cank, Giada Spigolon, Nicholas H. Oberlies, Bridget M. Barker, Jason E. Stajich, Christopher E. Mason, Kasthuri Venkateswaran

**Affiliations:** 1Jet Propulsion Laboratory, California Institute of Technology, Pasadena, CA 91109, USA; ceth.w.parker@jpl.nasa.gov (C.W.P.); nitin.k.singh@jpl.nasa.gov (N.K.S.); 2Pathogen and Microbiome Institute, Northern Arizona University, Flagstaff, AZ 86011, USA; marcus.teixeira@gmail.com (M.d.M.T.); Bridget.Barker@nau.edu (B.M.B.); 3School of Medicine, University of Brasilia, Brasilia 70910-900, Brazil; 4Department of Chemistry and Biochemistry, University of North Carolina at Greensboro, Greensboro, NC 27412, USA; haraja@uncg.edu (H.A.R.); k_cank@uncg.edu (K.B.C.); n_oberli@uncg.edu (N.H.O.); 5Biological Imaging Facility, California Institute of Technology, Pasadena, CA 91125, USA; giadas@caltech.edu; 6Department of Microbiology and Plant Pathology, University of California—Riverside, Riverside, CA 92521, USA; jason.stajich@ucr.edu; 7WorldQuant Initiative for Quantitative Prediction, Weill Cornell Medicine, New York, NY 10065, USA; christopher.e.mason@gmail.com

**Keywords:** biofilm, fungi, genomics, mars 2020 mission, metabolomics, morphological analysis, phylogenetic analysis

## Abstract

A fungal strain (FJII-L10-SW-P1) was isolated from the Mars 2020 spacecraft assembly facility and exhibited biofilm formation on spacecraft-qualified Teflon surfaces. The reconstruction of a six-loci gene tree (ITS, LSU, SSU, *RPB1* and *RPB2*, and *TEF1*) using multi-locus sequence typing (MLST) analyses of the strain FJII-L10-SW-P1 supported a close relationship to other known *Parengyodontium album* subclade 3 isolates while being phylogenetically distinct from subclade 1 strains. The zig-zag rachides morphology of the conidiogenous cells and spindle-shaped conidia were the distinct morphological characteristics of the *P. album* subclade 3 strains. The MLST data and morphological analysis supported the conclusion that the *P. album* subclade 3 strains could be classified as a new species of the genus *Parengyodontium* and placed in the family Cordycipitaceae. The name *Parengyodontium torokii* sp. nov. is proposed to accommodate the strain, with FJII-L10-SW-P1 as the holotype. The genome of the FJII-L10-SW-P1 strain was sequenced, annotated, and the secondary metabolite clusters were identified. Genes predicted to be responsible for biofilm formation and adhesion to surfaces were identified. Homology-based assignment of gene ontologies to the predicted proteome of *P. torokii* revealed the presence of gene clusters responsible for synthesizing several metabolic compounds, including a cytochalasin that was also verified using traditional metabolomic analysis.

## 1. Introduction

NASA microbial burden assessment of the spacecraft-associated surfaces is biased toward detecting endospore-forming bacteria as a primary planetary protection (PP) concern [1,2]. The extreme hardiness of bacterial endospores allows them to tolerate inhospitable conditions for long periods, making them particularly good candidates for surviving the journey to planetary bodies that may support life [3]. However, while fungal species also produce protective structures (spores, conidia, or cysts) as both part of their life cycle and as a response to environmental stress, few studies have examined their presence on the spacecraft-associated surfaces or their survival under simulated space conditions [4,5]. As a result, several reports on the description of novel bacterial species associated with spacecraft environments were published [6]. Still, systematic characterizations of fungal strains associated with spacecraft environments for their phylogenetic novelty are yet to be conducted.

In an ongoing microbial surveillance study of NASA Mars 2020 mission-associated spacecraft assembly environments, a novel fungal strain (FJII-L10-SW-P1) belonging to the genus *Parengyodontium* was isolated. The internal transcribed spacer (ITS) region-based phylogenetic analysis demonstrated that the Mars 2020 strain (FJII-L10-SW-P1) and three other isolates (LEC01, CBS 368.72, and UAMH 9836) were affiliated with the *Parengyodontium album* subclade 3. Subsequent whole-genome sequencing (WGS) analysis revealed that the strain FJII-L10-SW-P1 exhibited a strong phylogenetic relationship with the strain LEC01 that was isolated from a hydrocarbon gas turbine fuel sample, which was misidentified as *Lecanicillium* sp. using 18S rRNA gene phylogeny [7]. *P. album* strains were also isolated from a variety of ecosystems, including marine sediments [8], plant materials [9], soil [10], and walls/paintings [11,12].

The taxonomy of *Parengyodontium* is complex, as its members were originally assigned to the genus *Beauveria* [13], then to *Tritirachium* [14], and as *Engyodotium* [15]. Finally, phylogenetic analyses targeting the ITS region, 28S nuclear ribosomal DNA, and β-tubulin gene as well as matrix-assisted laser desorption ionization–time of flight mass spectrometry (MALDI-TOF-MS) profiles, resulted in transferring members of the *Engyodotium* species to a novel genus, *Parengyodontium*, within the family Cordycipitaceae [16]. At the time of writing, the genus *Parengyodontium* consists of *P. album* [16] and *P. americanum* [17]. The ITS-based phylogenetic analysis and MALDI-TOF profiles of several *P. album* strains displayed three distinct subclades, whereas the 28S rDNA-based phylogeny could not separate subclades 1 and 2 [16]. The cryptic species associated with subclades 1 and 2 need more study, but during this study, the WGS-based phylogeny and multi-locus sequence type (MLST) analyses revealed that strains belonging to subclade 3 should be classified as a novel species of the genus *Parengyodontium*.

The formation of microbial biofilms on surfaces, with consequent biofouling/biocorrosion of space hardware and life-support systems, is a significant concern to NASA and will also be of interest to commercial companies. In addition, the biofilm-suppressing materials will be helpful to several industries, including the health, medical instruments, oil, and water pipe industries. Hence, one of the objectives of this study is to isolate fungi from Mars 2020 assembly facility cleanroom and study on their biofilm formation by the Mars 2020 strain (FJII-L10-SW-P1) on space-qualified material surfaces. Although it is extremely unlikely to find significant biomass (much less biofilms) on flight hardware associated with robotic exploration, many of the materials used to fabricate robotic systems are also used in the construction of crewed spaceflight hardware. Thus, an attempt was made to understand whether a commercially available antimicrobial compound coated on metal surfaces or Teflon materials could resist biofilm formation by this novel fungal strain. The second objective of this study is to define the phylogenetic placement of the NASA Mars 2020 strain (FJII-L10-SW-P1) using microscopy and taxonomic affiliation based on MLST analyses, including six-loci (ITS, LSU, SSU, *RPB1* and *RPB2*, and *TEF1*) [18]. The third objective is to compare the WGS of FJII-L10-SW-P1 strain with closely related species and other cordycipitaceous fungi, and annotate the genomes using various bioinformatics pipelines, which might aid in the identification of genetic determinants related to, for example, biofilm formation and survival in harsh extraterrestrial environments. Furthermore, we predicted a wide range of secondary metabolite clusters from the genomes that are biotechnologically relevant, and fungal metabolites were also confirmed with metabolomics approaches that are common to the field of natural products research [19,20].

## 2. Materials and Methods

### 2.1. Sample Collection

In general, NASA cleanroom facilities are maintained with cleaning regimens at frequencies appropriate to the current level of activity in each cleanroom. For example, during the sampling at the Jet Propulsion Laboratory (JPL) spacecraft assembly facility (SAF) on 25 September 2018, a significant amount of assembly activity of critical Mars 2020 spacecraft hardware was present in the cleanroom; therefore, JPL-SAF was cleaned daily, including vacuuming and mopping with a cleaning solution (Kleenol 30, Cleancraft Industries, Inc., Commerce, CA, USA). Daily cleaning regimens include replacing tacky mats, wiping surfaces, and vacuuming/mopping floors using cleanroom-certified sanitizing agents (disinfectants, alcohol, or ultrapure water). All personnel who enter these cleanrooms must follow good manufacturing practice procedures to minimize the influx of particulate matter. Specific entry procedures vary depending on the certification level of the cleanroom and the presence or absence of mission hardware. General precautions include the donning of cleanroom-certified garments to minimize exposure of skin, hair, and the regular clothing of technicians and engineers. In addition, the use of cosmetics, fragrances, body spray, and hair gels were prohibited before entry into the cleanroom. Additionally, the air supplied to both facilities was filtered through high-efficiency particle arrestance (HEPA) filters.

Samples were collected from the cleanroom with 12 inch × 12 inch premoistened polyester wipes (Sterile TexTra10 TX3225; Texwipe, Kernersville, NC, USA) from 10 different locations (1 m^2^ each) as previously reported [21]. After sampling, polyester wipes were placed in a 500 mL bottle containing 200 mL of sterile PBS and vigorously shaken for one minute to dislodge microbial cells. These environmental samples were then concentrated using a CP-150 InnovaPrep concentrating pipette (Innova Prep LLC, Drexel, MO, USA) to a final volume of ~6 mL [22].

### 2.2. Isolation of Fungi

Most fungal species resist chloramphenicol, and hence it is used to suppress bacterial proliferation and allow for the isolation of fungi. Therefore, aliquots of the concentrated samples as mentioned above were treated with chloramphenicol (100 µg/mL) and incubated overnight at 25 °C. After 18 to 24 h of incubation, both chloramphenicol-treated and untreated samples were processed for the isolation of fungal species [23]. Subsequently, samples enriched in chloramphenicol for overnight incubation were 10-fold diluted, and 100 μL was added in duplicate to potato dextrose agar (PDA, Difco, Thermo Fisher Scientific, Irwindale, CA, USA) containing chloramphenicol (25 mg/L) and grown at room temperature (~25 °C). After 7 days, 75 of the colonies that grew on PDA were collected and stored as stab cultures and glycerol stocks for further analysis.

### 2.3. Morphological Analysis

For phenotypic/morphological characterization, the fungal strain was transferred to PDA and oatmeal agar (OMA, Difco), incubated at room temperature at 23 °C; colony size (in mm), structure, pigmentation, and characteristics were recorded after 21 days. PDA was utilized to determine microscopic traits, and cultures were allowed to grow for 7–9 days. The slide culture technique [24] was utilized to observe the microscopic morphology of the fungal strain. Briefly, a small block of agar was placed in the center of a sterile slide, and all four sides of the agar were inoculated with the fungus. Subsequently, a sterile cover slip is gently placed on the top of the block. The slide was kept in a moist chamber, made of a Petri dish lined with filter paper soaked in sterile water. After 3–4 days, the fungus grew out on the coverslip as well as the slide. The cover slip was gently picked up with sterile forceps and placed on a clean slide with a drop of water for observing details of conidiophores, conidia, and other microscopic structures, such as the width of hyphae. Photomicrographs were captured using phase and Nomarski contrast on an Olympus BX53 microscope with Olympus DP25 camera and Olympus cellSens software Version 1.7. Measurements of micromorphological characters were made with the Olympus cellSens software. Photographs of the colonies were taken with a Canon Power shot SD1300 IS.

### 2.4. Scanning Electron Microscopy

Following fresh fungal sample collection, cells were immersed in chilled 2.5% glutaraldehyde (Ted Pella Inc.; Redding, CA, USA) in 0.1 M sodium cacodylate buffer (Sigma–Aldrich, St. Louis, MO, USA) and incubated at 4 °C for 1 h before being washed 3 times in 0.1 M sodium cacodylate buffer. Cells then underwent isopropyl alcohol (IPA) dehydration via a series of incremental IPA steps from 50% to 100% (50%, 70%, 80%, 90%, 95%, and 3 times 100%) and stored at 4 °C in 100% IPA. Samples were critically point dried in an Automegasamdri 915B critical point dryer (Tousimis, Rockville, MD, USA). Samples were attached to scanning electron microscopy (SEM) stubs with carbon tape (Ted Pella Inc., Redding, CA, USA), followed by carbon coating with a Leica EM ACE600 Carbon Evaporator (Leica, Wetzlar, Germany) to a thickness of ~12 nm. SEM analysis was performed with an FEI Quanta 200F scanning electron microscope (Thermo Fisher, Waltham, MA, USA).

### 2.5. Biofilm Formation

Commercially available and patented organosilane, a surface-penetrating compound, was used as an antimicrobial coating on the tested surfaces. The active ingredient in the antimicrobial compound tested is 3-(trihydroxysilyl) propyldimethyloctadecyl ammonium chloride and previously assessed to control bacterial biofilm formation [25]. In the presence of hydroxyl groups at the surface of the glass, minerals, or metals (e.g., aluminum, steel), silanols formed a stable Si bond. This chemistry allowed silanes to function as valuable surface-treating/protecting or coupling agents. In addition, the antimicrobial compound breaks down the interfacial tension on surfaces, permitting the active ingredient to covalently bond (non-polar covalent bond) more quickly and evenly, resulting in better efficacy and protection. The spacecraft qualified materials tested during this study were purchased and precision cleaned at the JPL following NASA standard practices developed for cleaning spacecraft components, as previously described [26]. This cleaning step assured sterility, as no microorganisms were grown after cleaning when test coupons were placed in sterile nutrient media [26]. Once precision cleaned, the antimicrobial compound was applied and used.

Suitable quantities (10^6^ conidia) of conidial suspension were added to 10 mL of potato dextrose broth (PDB, Difco) in 50 mL polypropylene conical vials. Sterile Centers for Disease Control and Prevention (CDC) circular disc coupons (12.5 mm diameter × 1 mm thickness; BioSurface Technologies Corporation, Bozeman, MT, USA) composed of Inconel (aerospace nickel alloy) and Teflon were added separately to vials to serve as biofilm substrate, along with uninoculated controls. In addition, treated Teflon coupons coated in an antimicrobial coating were also added to separate vials to serve as substrates to test biofilm mitigation. All experiments were carried out in triplicate but SEM studies were conducted from a single coupon. Vials containing Inconel or Teflon were incubated at 25 °C in an orbital shaker at 25 RPM for at least 21 days prior to harvesting and analysis. Additionally, 10 mL of PDB was added to microscope compatible Nunc Glass Bottom Dishes (150680, Thermo Fisher Scientific), inoculated with fungal suspension, and incubated for 21 days. After suitable incubation periods, the coupons were tested for biofilm formation using confocal microscopic analyses.

### 2.6. Confocal Microscopy

Biofilm samples were analyzed via confocal microscopy. CDC coupons were removed from growth vials and immediately submerged into 4% paraformaldehyde (PFA; Sigma–Aldrich) in 1 × phosphate-buffered saline solution (PBS; Sigma–Aldrich) in a 24-well COTS plate. Coupons were then rinsed in PBS 3 times to ensure the removal of PFA. The cell walls of the samples were first stained with 15 µM Calcofluor White (Sigma–Aldrich) for 1 h at 37 °C, followed by rinsing with deionized-H_2_O 3 times. Samples were then stained with 1 µM TO-PRO-3 (Sigma–Aldrich) for nucleus staining at room temperature for 30 min, followed by rinsing with deionized-H_2_O for 3 times. Samples were then stored at 4 °C. Samples were protected from light throughout the staining process by wrapping the 24-well COTS plate with aluminum foil.

Confocal imaging was performed on ZEISS LSM 980 with Airyscan 2 (Zeiss, Jena, Germany) at the California Institute of Technology Biologic Imaging Facility using a Zeiss 63X Plan-Apochromat 1.4 NA (Zeiss). Image processing and analysis were performed with the Imaris software package (Version 9.01.05). Surface filling was used to generate a composite of TO-PRO-3 and Calcofluor White model. Due to the auto-fluorescence of both the background Teflon surface and the coverslip, both surfaces were cropped out of the final confocal image to provide higher clarity and more accurate analysis. The biofilm density was quantified by determining the number of voxels per Z slice that contained the surface-filling model. We used these data to determine the median height of each biofilm. The proportionate biomass within each biofilm was calculated by multiplying the number of voxels containing surface-filled biofilm by the voxel size (0.132 µm L × 0.132 µm W × 1.25 µm H). 

### 2.7. ITS-Based Fungal Identification

Among the 75 fungal isolates, 12 strains were novel based on ITS-sequence analyses; however, we have performed detailed phylogenetic analyses only for the FJII-L10-SW-P1 strain due to its capability in forming biofilm on spacecraft qualified materials. DNA was extracted from all fungal strains (n = 75 isolates) using the Maxwell-16 MDx automated system following the manufacturer’s instructions (Promega, Madison, WI, USA). Initial identification of the fungus was performed by amplicon sequencing targeting the internal transcribed spacers (ITS) region using primers ITS 1F (5′-CTT GGT CAT TTA GAG GAA GTA A-3′) [27] and Tw13 (5′-GGT CCG TGT TTC AAG ACG-3′) [28]. PCR conditions and sample preparation steps for sequencing were performed as described elsewhere [29].

The ITS sequences were characterized through the Basic Local Alignment Search Tool (BLAST) algorithm [30] using the National Center for Biotechnology Information (NCBI) and UNITE database(s) to find the type strains with the closest percent similarity to the fungal strain. Sequences from all taxa were obtained from the two previous taxonomic studies on the genus *Parengyodontium* [16,17] and sequence data for other closely related taxa from the family Cordycipitaceae were downloaded from NCBI. The ITS dataset comprising 24 sequences was used for an initial phylogenetic analysis, among which 17 sequences belonged to the genus *Parengyodontium*. The sequences were aligned using ClustalW followed by generation of the Maximum-Likelihood (ML) tree using MEGA 7.0.26 [31]. One thousand bootstraps were performed to test branch fidelity.

### 2.8. MLST-Based Phylogenetic Analyses

Two different MLST analyses were utilized due to a lack of resolution from the ITS region alone to resolve the phylogenetic affiliations of certain fungi and are detailed below. Since the number of genetic markers are uneven among Cordycipitaceae taxa (especially beta-tubulin), we decided to split the MLST scheme into two concatenated datasets, as follows:(a)Three-gene MLST analyses. Sequences from ITS, 28S nrDNA, and β-tubulin genes were used in a dataset comprised of 22 fungi, including the outgroup. The outgroup selection was based on [16]. Multiple sequence alignments were generated using MAFFT default settings using PhyloSuite v.1.2.1 [32]. The alignments were trimmed to remove ambiguous characters using GBlocks [33,34]. For the concatenated dataset, PartitionFinder 2 [35] was used to select the best-fit model according to the Akaike Information Criterion corrected (AICc) [36]. The best-fitting substitution models according to AICc were: ITS and β-tubulin: GTR+I+G and LSU: TRN+I. ModelFinder [37] was used for the ITS dataset to select the best-fit model using the AICc criterion. The best-fit model according to AICc was TIM2+F+R2. The trimmed alignment was then used to construct a ML tree using IQ-TREE implemented in PhyloSuite. Ultrafast bootstrapping was done with 5000 replicates [38]. Nodes with UFBoot ≥90% are shown on the clades, but only nodes ≥95% were considered strongly supported. Bayesian inference phylogenies were inferred using MrBayes 3.2.6 [39] under partition model (2 parallel runs, 10 million generations), using PhyloSuite v. 2.1. Four independent chains of Metropolis-coupled MCMC were run for 10 million generations with trees sampled every 1000th generation, resulting in 10,000 trees. The first 25% of the trees were discarded as a burn-in parameter. The average standard deviation of split frequencies value approaching 0.001 was used to estimate that the two runs had converged closer to the stationary phase (10 million generations). Consensus trees were generated and viewed in PAUP* v.4.0a (build 166) [40]. Clades with a posterior probability (PP) ≥95% were considered significant and strongly supported.(b)Six-loci MLST analyses. Gene sequences utilized were: ITS region rRNA gene, D1/D2 domain of large subunit (LSU or 26S) rRNA gene, small subunit (SSU or 18S) rRNA gene, and housekeeping genes including two subunits of RNA polymerase II (*RPB1* and *RPB2*) and the translation elongation factor 1-α (*TEF1*). These six-loci have already been established for differentiating Cordycipitaceae species [41]. Sequences of 58 fungal strains available were downloaded, and sequences were manually concatenated (representative sequences are available at [41]. The respective gene sequences that were available on NCBI for different *Parengyodontium* species (n = 8 isolates) were included in the phylogenetic analysis except for the Mars 2020 strain (FJII-L10-SW-P1), which was generated during this study. For MLST, sequences were aligned using MAFFT v7 [42], concatenated manually, trimmed using the ClipKit tool, smart-gap function [43] and a ML Tree was generated using the using IQTREE2 v2.0.6 [31,44]. The best substitution model was calculated using the ModelFinder algorithm [37] and 1000 ultrafast bootstraps [45] and SH-like approximate likelihood ratio test (aLRT) were used to test branch support [46]. Finally, the trees were visualized using the FigTree v 1.4.4 software (http://tree.bio.ed.ac.uk/software/figtree/, accessed on 12 November 2021).

### 2.9. Whole-Genome Sequencing Analyses

A pure and well-isolated colony was picked after streaking onto PDA plates, and approximately 1-g wet weight mycelia were collected for DNA extraction. DNA extraction, WGS, and processing were followed, as described elsewhere [47]. Briefly, the total nucleic acid extraction was carried out using ZymoBIOMICS 96 MagBead DNA kit (Lysis tubes) (Zymo Research, Irvine, CA, USA) after bead beating with Precellys homogenizer (Bertin, Rockville, MD, USA). This was followed by library preparation using the Illumina Nextera Flex Protocol as per Illumina document number 1000000025416 v07. The initial amount of DNA for library preparation was quantified, and 5 to 12 cycles of PCR were carried out to normalize the output depending on the input DNA concentration. The amplified genomic DNA fragments were indexed and pooled in a 384-plex configuration with dual-index adapters. Whole-genome shotgun sequencing was performed on a NovaSeq 6000 S4 flowcell PE 2 × 150 platform with a paired-end module. The data was filtered with NGS QC Toolkit v2.3 [48] for high-quality (HQ) vector and adaptor-free reads for genome assembly (cutoff read length for HQ, 80%; cutoff quality score, 20). The number of filtered reads obtained were used for assembly with SPAdes 3.14.0 [49] genome assembler (k-mer size: 32 to 72 bases) using default parameters. The resulting assembly was curated next using the AAFTF pipeline [50], as follows: (1) Mitochondrial and contaminant contigs were identified and removed with the “vecscreen” and “sourpurge” functions; (2) the “rmdup” function was used to remove duplicated contigs identified by the minimap2 [51] algorithm; (3) the final assembly was polished using the “pilon” function in order to correct bases, fix misassembled contigs and fill potential gaps [52]; (5) the contigs smaller than 1000 bp were purged and sorted by size using the “sort” function; (6) finally, the assembly statistics were obtained using the “assess” tool. The genome completeness of the FJII-L10-SW-P1 assembly was evaluated using the BUSCO v5.1.2, sordariomycetes_odb10 library [53]. Genomic DNA sequences of all available strains used in this study were downloaded from NCBI, and comparative analyses were performed (Appendix A).

### 2.10. De Novo and Functional Genome Annotation

The resulting assembly of the FJII-L10-SW-P1 strain was annotated using the funannotate v 1.8 pipeline, called the “annotate” function [54]. We re-annotated the genomes of other Cordycipitaceae fungi (Table 1 and Appendix A) in order to avoid annotation bias for comparative genomic analysis. Initially, repetitive sequences were soft-masked using the tantan algorithm [55]. We generated and curated the library of repeats identified de novo by RepeatModeler v2.0.1 [56] combined with well-characterized transposable elements from RepBase [57] for fungi. The genome was masked with RepeatMasker v4.0.7 [58] to identify these interspersed repeats based on the library and low complexity DNA sequences. Gene prediction using ab initio gene predictors was performed using the funannotate “predict” function on the masked genome; GeneMark-ES v4.62 was used to predict genes using the self-training algorithm [59] while AUGUSTUS v3.3.3 [60], Glimmerhmm v3.0.4 [61], and SNAP v 0.15.4 [62] predicted the gene structures based on a combination of high-fidelity transcripts of *Beauveria bassiana* ARSEF 2860 and *Cordyceps militaris* CM01 as well as using the BUSCO lineage-specific sordariomycetes_odb10 library [53]. The consensual gene prediction was achieved using the EVidence Modeler (EVM v1.1.1), where a weight of 2 was attributed to high quality AUGUSTUS predictions, while in the other ab initio predictions, the weight was set to 1 [63]. Gene models with less than 50aa, spanning gaps, or containing transposable elements were removed. The tRNAs were identified using tRNAscan-SE v 1.3.1 [64]. Genome annotation was performed using the “annotate” function of funannotate. This was achieved with searches of the predicted proteins by diamond blastp [65] to the UniProt DB version 2021-03. Further alignments to Pfam v31.0 [66] and InterPro5 [67] domains, carbohydrate-active enzymes (CAZymes) [68], secreted proteins [69], transmembrane proteins [70], proteases (MEROPS) [71], fungal transcription factors [67], and BUSCO groups [53] were found. The GO terms were assigned based on matches to the InterPro database searches. Secondary metabolite gene clusters were identified with antiSMASH v6.0.0 [72]. Matches to Eggnog and the cluster of orthologous groups of proteins (COGs) orthologous groups were further added to the functional annotation. The functional descriptions from UniProtKB/SwissProt best matches at 80% alignment and 80% identity were combined with descriptions from Eggnog-mapper searches to generate gene names and product descriptions. The mitogenome was annotated using the RNAweasel and MFannot pipelines (https://github.com/BFL-lab/Mfannot, accessed on 12 November 2021). Lastly, the annotations collected from each genome were converted into the GenBank flat-file format (gbk) for comparative genomic analysis, and the generated.tbl and. sqn files were submitted to NCBI Genomes. 

## 3. Comparative Analysis of Fungal Genomes

Comparative genomic analyses were performed using the “compare” function of funannotate pipeline using nine Cordycipitaceae fungi. Initially, to generate a phylogenomic informed species tree, a set of single-copy orthologs between the nine genomes analyzed were identified using the Proteinortho v6.0.20 [73]. Individual single-copy orthologs were aligned using the MAFFT v7 pipeline [42] and trimmed using the ClipKit tool and smart gap function [43]. Therefore, each unique alignment was submitted for ML analysis using the IQTREE2 software [44]. We set the species *Simplicillium aogashimaense* 72–15.1 as the outgroup, and each individual best protein substitution model was calculated using the ModelFinder method [37]. The concatenated tree and individual trees were submitted to branch fidelity using two different approaches in IQTREE2 software: Ultrafast bootstraps [45] and Gene Concordance Factors [74]. The phylogenomic tree was visualized using the FigTree v1.4.4 pipeline (http://tree.bio.ed.ac.uk/software/figtree/, accessed on 12 November 2021).

The copy number counts Pfam, CAZymes, MEROPS, transmembrane proteins, secreted proteins, COGs, secondary metabolites, and fungal transcription factors and plotted the categories with a standard deviation >1 in a heat map. We also compared the GO-enriched terms for each species by taking into account those with an FDR GO-enrichment *p*-value < 0.05. To identify specific GO categories enriched for the *Parengyodontium* lineage as well for each individual of this genus, we used the OrthoVenn2 approach [75]. The predicted proteomes from each individual were submitted to OrthoVenn2 for visualization and comparison of gene content, and the enriched GO categories with a *p*-value < 0.05 were retrieved. We also looked for the presence of genes related to biofilm formation and adhesion, melanin biosynthesis, radioresistance, and microgravity resistance in the genome of the strain FJII-L10-SW-P1.

### Metabolomics

The cultures of fungal strain FJII-L10-SW-P1 were maintained on potato dextrose agar (PDA; Difco). An agar plug from the leading edge of the PDA culture was transferred to a sterile tube with 10 mL of liquid PDA. The culture was grown for 12 days on an orbital shaker (100 rpm) at room temperature (rt; ~23 °C) and then used to inoculate the solid fermentation media, as described below. Solid-state fermentations were carried out in 250-mL Erlenmeyer flasks. To prepare the two media, 10 g of rice and oatmeal were added to separate flasks (two rice and two oatmeal flasks) with 50 mL of deionized water. After autoclaving these samples at 120 °C for 20 min, the flasks were inoculated with the fungal culture (10 mL of 7 day grown cultures) to be tested and incubated at room temperature for 2 weeks. Subsequently, each of the two solid-state fermentation cultures of FJII-L10-SW-P1 were chopped up into small pieces using a spatula, and 60 mL of 1:1 MeOH-CHCl_3_ were added. The fungal cultures were then shaken using a rotary shaker overnight (~16 h) at ~125 rpm at room temperature. The cultures were filtered in vacuo and then pooled to form a combined filtrate, and the solid residue was rinsed with a small volume of 1:1 CH_3_OH-CHCl_3_. To the filtrate, 90 mL of CHCl_3_ and 150 mL of H_2_O were added; the solution was stirred for 20 min and transferred to a separatory funnel. The organic layers were collected and evaporated to dryness under vacuum using a rotary evaporator. The resulting organic layers were partitioned between 100 mL of 1:1 CH_3_OH-CH_3_CN and 100 mL of hexane. The CH_3_OH-CH_3_CN layers were collected and evaporated to dryness under vacuum to yield two organic extracts, one from rice (77.12 mg) and one from oatmeal media (253.87 mg).

High-resolution electrospray ionization mass spectrometry (HRESIMS) was performed on a Thermo LTQ Orbitrap XL mass spectrometer (Thermo Fisher, San Jose, CA, USA) equipped with an electrospray ionization source. Source conditions in the positive-ionization mode were set at 275 °C for the capillary temperature, 4.5 kV for the source voltage, 20 V for capillary voltage, and 95 V for the tube lens. Nitrogen was utilized for the sheath gas and set to 25 and 20 arb for the positive and negative modes, respectively. For the negative-ionization mode, nitrogen was also used as an auxiliary gas and set at 10 arb. Scan events were carried out, with full-scan (*m*/*z* of 100–2000) and ion-trap MS/MS of the most intense ion from the parent mass list utilizing CID with a normalized collision energy of 30. Thermo Scientific Xcalibur 2.1 software was used for instrument control and data analysis. Ultra-performance liquid chromatography (UPLC) was carried out on a Waters Acquity system using a BEH C18 (2.1 × 50 mm, 1.7 μm) column (Waters Corp., Milford, MA, USA) equilibrated at 40 °C. A mobile phase consisting of CH_3_CN–H_2_O (acidified with 0.1% formic acid) was used, starting with 15:85 then increasing linearly to 100% CH_3_CN within 8 min, holding for 1.5 min, and then returning to the starting conditions within 0.5 min. An Acquity UPLC photodiode array detector was used to acquire PDA data, which were collected from 200 to 500 nm with a 4 nm resolution.

## 4. Results

### 4.1. Taxonomy of the Strain FJII-L10-SW-P1

*Parengyodontium torokii*, N.K. Singh and K. Venkateswaran, sp. nov.

**MycoBank number**: MB841139.

**Etymology**: Torokii refers to name Dr. Tamas Torok, an American mycologist conducting research on extremophiles).

**Diagnosis**: Similar to *Parengyodontium album* but phylogenetically unique and morphologically distinguished by its subcylindrical to ellipsoidal conidia.

**Holotype:** USA: Pasadena, CA, 34.1478° N, 118.1445° W, JPL-SAF cleanroom floor where the Mars 2020 mission components were assembled, 25 September 2018. Nitin K. Singh and Kasthuri Venkateswaran, (HOLOTYPE is stored in a metabolically inactive state as a lyophilized culture at the Northern Regional Research Laboratory [NRRL], Agricultural Research Service, USA; ex-holotype culture, FJII-L10-SW-P1 = NRRL 64203, conidial isolate from HOLOTYPE). GenBank accession numbers of the type strain (FJII-L10-SW-P1): ITS  = MT704894, Draft genome = JADQAY000000000. The genome size is 30.4 Mbp and the G + C content is 50.45 mol%.

**Description**: Colonies on PDA and OMA are white, floccose, cottony, velvety, opaque, with colorless exudates on the colony surface of PDA without diffusible pigments, and are reverse pale yellow with or without ridges (Figure 1A,B). After three weeks incubation at room temperature (23 °C), colonies on PDA have a 40 mm diameter, and colonies on OMA have a ~50 mm diameter. Vegetative hyphae are smooth-walled, hyaline, and septate. Conidiophores are erect, arising from hyphae at right angles, tapering to subcylindrical, slightly swollen at base, occasionally biverticillate, and bearing one to numerous whorls of conidiogenous cells (one to ≥five). Basal portion of the conidiogenous cell is elongated, tapering, 16–22 µm × 1–2 µm, terminating in fertile zigzag-shaped rachides, and bearing conidia (Figure 1C). Conidia are one-celled, smooth, thin-walled, hyaline, sub cylindrical to ellipsoidal, aseptate, apiculate, 2–3 µm × 1–2 µm, and arising from alternating points with butt-shaped denticles on zigzag-shaped and genticulate rachides (Figure 1D). No sexual state observed.

**Ecology/Substrate/Host**: Cleanroom floor where spacecraft components are assembled. 

**Other materials examined**: Three other strains belong to *P. album* subclade 3 CBS 368.72, UAMH 9836, and LEC01 were isolated from turbine fuel sample, Dayton, Ohio, USA (LEC01); from fresco, Romania (CBS 368.72); and from a human bronchoscopy specimen, Canada.

**Notes:** The new species, *P. torokii* is both morphologically and phylogenetically unique from other described members of the genus *Parengyodontium* [16,17]. Phenotypically, *P. torokii* can be readily distinguished from *P. album* based on conidial shape. *P. torokii* produces subcylindrical to ellipsoidal conidia, whereas *P. album* conidia are globose, smooth, hyaline, oval, and apiculate [16]. *P. torokii* differs from *P. americanum* as the former produces terminal fertile zigzag shaped rachides, but the latter lacks them and produces conidia on right-angled phialides or aphanophialides. In addition, the conidia of *P. torokii* differ from those of *P. americanum* in that they are sub cylindrical to ellipsoidal vs. cylindrical to globose in *P. americanum* [17]. Interestingly, the strain CBS 368.72 (subclade 3, sensu Tsang [16]) was morphologically similar to *P. torokii* in conidial shape and zigzag rachides based on SEM (Appendix A). Based on Maximum Likelihood molecular phylogenetic analyses of the ITS region (Appendix A) as well as three loci analysis (Appendix A), *P. torokii* is a distinct species as it occurs on a unique clade (subclade3 sensu Tsang [16], Appendix A). Further, in the six-loci analysis, *P. torokii*, *P. album*, and *P. americanum* are seen as distinct clades with moderate to significant statistical support (see below for details).

### 4.2. Biofilm Formation of the Strain FJII-L10-SW-P1

The SEM of vegetative cells of the FJII-L10-SW-P1 strain revealed thin membranous white layers surrounding the conidia and were presumed to be composed of molecules such as extracellular polymeric substances (EPS), which enabled biofilm formation (Figure 1E and Appendix A). When the biofilm formation was characterized on three different materials as substrate, the FJII-L10-SW-P1 strain was able to form biofilms on the Teflon (tetrafluoroethylene) coupons as well as on the plastic (polypropylene) walls of the conical Falcon tubes and the plastic (polystyrene) of the sides of the glass (borosilicate) bottomed Petri dish. Confocal imaging of the Teflon coupons (Figure 2) indicated that there was biomass present on both the uncoated and the antimicrobial-coated coupons. The uncoated coupon had patches of high-density biofilm and regions of no fungal mycelium, while the antimicrobial coated coupon had a much more distributed density of mycelium across the area that was imaged. Both antimicrobials coated and uncoated coupons showed lower amounts of biofilm at the substrate surface of the coupon (purple and blue colors, Figure 2) and more biofilm near the top of the biofilm (orange and red colors). Quantification of surface-filling voxels indicated that the biofilm formed on the uncoated Teflon is smaller (1767 µm^3^) and the median height is closer to the surface (94 µm) while the biofilm formed the coated Teflon coupon is larger (3326 µm^3^) and has a higher median height (107 µm). The shape of both biofilms resembles canopy morphology. Biofilms were not formed on either the glass-bottomed Petri dishes or on the Inconel coupons. Both borosilicate glass surfaces and Inconel (a nickel-chrome superalloy) are smooth surfaces, while Inconel is additionally resistant to corrosion and oxidation.

### 4.3. Phylogenetic Analyses of the Strain FJII-L10-SW-P1

Currently, the Mycobank and CBS databases documented only two *Parengyodontium* species, specifically *P. album* and *P. americanum.* The ITS sequences available on NCBI for *Parengyodontium* species including the strain FJII-L10-SW-P1 (n = 18 isolates) and other closely related species (n = 6) were used in the ML phylogenetic analysis with *Isaria coleopterora* (CBS 110.73) as an outgroup. A phylogram of the most likely tree (−lnL = 1622.37) from a ML analysis of 24 sequences based on the ITS region (568 bp) using IQ-TREE is shown in Appendix A. Among the 18 strains of *P. album* there were 3 subclades and to confirm that four strains that form the *P. album* subclade 3 belong to a novel species, their phylogenetic affiliations were analyzed. Next, a phylogram of the most likely tree (−lnL = 3824.45) from a 3-gene ML analysis of 22 sequences based on the combined regions of ITS, LSU, and β-tubulin gene (1435 bp) using IQ-TREE, was created (Appendix A). The 3-gene MLST phylogram also supported the phylogenetic clusters that was noticed in the ITS-tree, forming a separate branch for four strains including FJII-L10-SW-P1 isolate, which was distinct from *P. album* subclade 1 and 2.

Subsequently, a six-gene MLST analysis was carried out by manually concatenating ITS, LSU, SSU, RPB1, RPB2, and TEF1 gene sequences. In this analysis, in addition to the Mars 2020 isolate FJII-L10-SW-P1, three other strains belong to *P. album* subclade 3 (CBS 368.72, UAMH 9836, and LEC01 isolates), two strains of *P. album* (HKU48 and IHEM 4198 isolates), two strains of *P. americanum* (AZ2 and CA11 isolates), one strain of *Lecanicillium kalimantanense* BTCC-F23 and one strain of *Torrubiella wallacei* CBS 101237 were included, and the tree was rooted with members of the genus *Simplicillium*. The 5-gene MLST analysis confirmed that the FJII-L10-SW-P1 strain and other three strains clustered in a single clade are distinct from *P. album* and *P. americanum* (Figure 3). Single loci phylogenetic analyses (for example ITS; Appendix A) always placed FJII-L10-SW-P1 strain as a sister cryptic species of *P. album*. This cluster is supported by 65.6/92% of bootstrap and aLRT analyses (Figure 3).

The phylogenetic and genetic distinctiveness and morphological characteristics were sufficient to categorize the four strains that belong to *P. album* subclade 3, as members of a species distinct from other recognized *Parengyodontium* species. Therefore, on the basis of the data presented, strains FJII-L10-SW-P1, CBS 368.72, UAMH 9836, and LEC01 represent a novel species of the genus *Parengyodontium*, for which the name *Parengyodontium torokii* sp. nov. is proposed. As all four *P. torokii* strains were isolated from various niches, the ecological importance of this novel *Parengyodontium* species will be significant and warrants further studies.

### 4.4. Whole-Genome Sequence Analyses

A phylogenomic approach based on 5334 single-copy orthologs screened in nine available whole genomes was carried out in order to confirm the phylogenetic placement of the *P. torokii*. By rooting the tree with *Simplicillium aogashimaense*, genomes of *P. torokii* FJII-L10-SW-P1 and LEC01 strains are clustered in a monophyletic branch, next to species *P. americanum* species. The *Parengyodontium* group is highly supported by bootstrap and gene concordance factors (100/96.2—Figure 4). Unfortunately, no genomes of *P. album* are available; this is sorely needed to precisely define the phylogenomic relationships, along with other Cordycipitaceae fungi, such as *L. kalimantanense* and *T. wallacei*.

### 4.5. Genomic Features of the P. torokii FJII-L10-SW-P1 Strain

The draft genome of *P. torokii* FJII-L10-SW-P1 strain was assembled into 440 scaffolds with a genome size of 30.4 Mbp (Table 1). The largest scaffold was assembled into 718,708 bp and other genome statistics, such as N50 (122,374), L50 (70) of the draft genome, are given in Table 1. The annotated genome is deposited and the accession number is JADQAY000000000. For comparative genomic analyses, the genomes of *P. torokii* (FJII-L10-SW-P1 and LEC01), *P. americanum* AZ2, and six other Cordycipitaceae members published elsewhere were included (Table 1). The GC content of the *P. torokii* was about 50.45%, whereas *P. americanum* was 52.88%. The assembled genome of *P. torokii* had 2.83% of repeated DNA and was masked for annotations. The transposable elements detected in the *P. torokii* FJII-L10-SW-P1 strain are listed in Appendix A. We identified a plethora of retroelements (Penelope, LINEs, CRE/SLACS, R1/LOA/Jockey, LTR elements, Ty1/Copia, and Gypsy/DIRS1) and DNA transposons (hobo-Activator, Tc1-IS630-Pogo, En-Spm, MuDR-IS905, PiggyBac, and Tourist/Harbinger). We have also identified rolling-circles, small RNAs, simple repeats, and low complexity sequences (Appendix A).

The final annotation of the *P. torokii* FJII-L10-SW-P1 strain yielded 9596 protein-coding genes and 70 tRNA genes. As observed in the *P. torokii* FJII-L10-SW-P1 strain, a similar number of protein-coding genes (9801) and tRNA genes (77) was noticed in the *P. torokii* LEC01 strain. By comparing the overall gene content with other Cordycipitaceae species, we observed that the *P. torokii* FJII-L10-SW-P1 strain lineage has a lower number of genes and the predicted average length of genes and proteins products are higher in the *Parengyodontium* lineage (Table 1). The mitochondrial genome of the *P. torokii* FJII-L10-SW-P1 strain was also assembled and annotated (Appendix A). The full mitogenome was assembled into a single circular contig harboring 27,039 bp. We identified 14 mitochondrial genes responsible for the assembly of ubiquinone oxidoreductase (*nad1*, *nad2*, *nad3*, *nad4*, *nad4L*, *nad5,* and *nad6*—complex I), cytochrome b (*cob*—complex III), cytochrome oxidase (*cox1*, *cox2,* and *cox3*—complex IV), and ATP synthase (*atp6*, *atp8,* and *atp9*—complex V), one ribosomal protein (rns), and 24 transfer RNAs.

The genomic annotations revealed that the number of secreted proteins is lower in the *Parengyodontium* lineage than other Cordycipitaceae species (Appendix A). The lower number of secreted proteins predicted is also reflected in the reduction of carbohydrate-degrading enzymes (CAZy; Figure 5B; Appendix A) and peptidases (MEROPS; Figure 5A). We observed that both the genomes of *P. torokii* FJII-L10-SW-P1 and LEC01 strains, along with *B. bassiana,* have lower CAZY and MEROPS classes compared to other fungi characterized during this study. Significant changes in the number of each group are highlighted in Figure 5. Detailed analyses observed that some specific families of those enzymes are expanded in the *Parengyodontium* group. For example, the CAZy Glycosyl Hydrolase GH33 family is unique for the *Parengyodontium* group of fungi not shared with other Cordycipitaceae species. This enzyme, also known as sialidases (E.C. 3.2.1.18), is responsible for hydrolyzing a wide variety of N-acetylneuraminic acids linked to various types of sugars. We also observed an increase of the CAZy Glycosyltransferases families GT20 and GT34 compared to its relative species shown in Figure 5B. More specifically, members of the GT20 family (E.C. 2.4.1.15) are involved in the trehalose biosynthesis, while GT34 members are involved in the galactomannan. Regarding the peptidases, we observed that the MEROPS metallopeptidase families M24B, M28E, and M35 are expanded in the *Parengyodontium* species compared to other species. Metalloproteases (E.C. 3.4.24.) are characterized by displaying a catalytic metal in the active site of the protein and a conserved consensus HEXXH motif. Lastly, the function of metalloprotease, belonging to the family M35, has been widely investigated in fungal genomes, and has been associated with facilitating the penetration of Sordariomycetes species in insect cuticles [76] or in plant hosts [77].

We also compared the profile of fungal-specific transcriptional factors among the Cordycipitaceae species (Figure 6). In general, we observed the enrichment of certain classes of transcriptional factors in the *Parengyodontium* lineage compared to other fungi. For example, the classes KilA-N domain (IPR018004), Basic region leucine zipper 2 (IPR004827), Helix-loop-helix DNA-binding domain (IPR011598), Myb-like DNA-binding domain (IPR001005), bZIP TF 1 (IPR004827), Fungal Zn(2)-Cys(6) binuclear cluster domain (IPR001138) and Fungal-specific TF domain (IPR007219) are enriched in this particular Cordycipitaceae group. It is worth noting that such enrichments are more prominent in *P. torokii* LEC01 and *P. americanum* AZ2 compared to *P. torokii* FJII-L10-SW-P1.

We identified genes related to biofilm formation, adherence, and glycosaminoglycan (GAG) biosynthesis in the *P. torokii* genome. From a set of 36 genes analyzed, we found 34 *A. fumigatus* homologues in the *P. torokii* FJII-L10-SW-P1 genome. These include glycosyl hydrolases (GHs) and glycosyl transferases (GTs) related to polysaccharide biosynthesis, transcription factors, genes belonging to the MAPK signaling pathway, and transmembrane transporters (Appendix A). As the FJII-L10-SW-P1 strain was able to tolerate harsh and inhospitable conditions during the NASA Mars 2020 mission, we looked for genes related to pigment (melanin) biosynthesis, radioresistance, and microgravity resistance well characterized in other fungal species (Appendix A). *P. torokii* FJII-L10-SW-P1 can produce melanin via L 3–4 dihydroxyphenylalanine (L-DOPA) and L-tyrosine pathways but not via the DHN-melanin pathway. For both L-DOPA and L-tyrosine we identified all homologs well characterized in *A. fumigatus* and *A. niger* (Appendix A). However, we did not find the *arp1* homologue that codifies for a protein containing a scytalone dehydratase domain. This enzyme is responsible for the production of 1,8-Dihydroxynaphthalene (1,8-DHN), which is a key intermediate DHN melanin pathway. We also found genes related to radioresistance; we found homologues of recombinases essential for repairing double-stranded DNA breaks caused by radiation. We also identified genes important for microgravity resistance in the *P. torokii* genome. Genes related to the biosynthesis of proteinase K, DNA repair protein rad9, and daughter-specific expression protein 2 were identified. We did not find homologues of flocculin, which is an important protein to tolerate microgravity conditions. This enzyme is not observed in filamentous fungi but in yeasts from the saccharomycotina subphylum (Appendix A).

Finally, we also looked for enriched GO terms in the *Parengyodontium* lineage within the *P. torokii* and *P. americanum* species. We used either *S. aogashimaense* (Figure 7A) or *L. fungicola* (Figure 7B) as outer species for the OrthoVenn comparisons with *P. torokii* FJII-L10-SW-P1, LEC 01, and *P. americanum* AZ2. We identified clusters of proteins shared by all four fungi and those shared within all *Parengyodontium* species and those specific for *P. torokii* and *P. americanum*. According to the enrichment analysis, we observed that the transmembrane transport category (Biological Process—GO:0055085) is enriched in the *Parengyodontium* lineage in both scenarios. For *P. torokii*, we identified that transmembrane transport (GO:005508), ATPase activity (GO:0042626) and oxidoreductase activity (GO:0016705) terms are enriched while transaminase activity (GO:0008483) is enriched in the *P. americanum.*

### 4.6. Metabolomic Profiling of P. torokii FJII-L10-SW-P1 Strain

To access the metabolic profile of the *P. torokii* FJII-L10-SW-P1, the fungal extracts that were generated from samples grown separately on rice and oatmeal media were analyzed by LC–MS to verify similarities between the identified gene clusters predicted by in silico genomic analysis and the fungal metabolite production of the fungus. The two extracts showed similarities in their metabolite production based on their based peak chromatogram and photodiode array detector data collected from the LC-MS analysis (Figure 8A–C). The extracted-ion chromatograms (XIC) were used to identify compounds that have previously been described by the same biosynthetic gene clusters (Figure 8C). We found that cytochalasin K was putatively biosynthesized by this fungus under these growth conditions (Figure 8B), as the accurate mass of cytochalasin K was found in both extracts within 5 ppm of the compound’s accurate mass value (*m*/*z* 532.2708 [M+H]^+^; calcd for C_32_H_38_NO_6_, *m*/*z* 532.26991; Figure 8C). This biosynthetic cluster was also identified by in silico analysis and is conserved among other Pezizomycotina fungi (Figure 8D). In addition, we identified 32 secondary metabolite clusters predicted by antiSMASH (Appendix A). From those, we identified a putative cluster related to the production of equisetin, cephalosporin C, EQ-4, squalestatin S1, curvupallide-B, pyranonigrin E, and dimethylcoprogen (Appendix A). In addition, using a metabolite identification protocol and an in-house database of over 650 fungal metabolites [19,20], we also note the level one identification [78] of the following metabolites: 5,8-Epidioxyergosta-6,9(11),22-trien-3-ol, cephalochromin, (E)-2,3-dihydroxypropyl octadec-6-enoate, cyclo(L-Leu-L-Pro), betulinan, 6,9-Octadecadienoic acid, ergosta-4,6,8(14),22-tetraen-3-ol, and (3β,22E)-cyclo (L-Pro-L-Leu) (Appendix A).

## 5. Discussion

The genus *Parengyodontium* was only recently proposed [16]; however, the fungi clustered within this group have been described and widely studied for several years [13,14,15]. The ecology of *P. album* and *P. americanum* suggests that these fungi are ubiquitous in the environment and are potential opportunistic fungal pathogens of humans [16,17]. In this communication, we described a novel *Parengyodontium* species that was isolated from the Mars 2020 spacecraft assembly facility, suggesting that they tolerate oligotrophic environments. Morphology and conidial architecture of *P. torokii* is unique from other validly described members of the genus *Parengyodontium*. The *P. torokii* strains produce subcylindrical to ellipoisdal conidia, whereas *P. album* conidia are globose, smooth, hyaline, oval, and apiculate [16]. *Parengyodontium americanum* conidia are also different from *P. torokii* by virtue of cylindrical to subglobose shape [17]. For more than a century, identification of fungi was carried out using microscopic observations of the sexual structure and conidial formation followed by colony morphology. As delimitation of microbial species has been notoriously challenging using traditional culture methods, the utilization of molecular approaches dramatically changed the taxonomy of fungi [79].

Initial strain characterization using the ITS region as a barcode identifier found that the four strains of *P. torokii* clustered into one group. However, it was reported that not all fungal phyla can be differentiated using the ITS region as a molecular barcode and remains simply as a first diagnosis [80]. Depending on the fungal genera, the use of MLST is strongly recommended to identify cryptic species [18] and can be combined with phenotypic and reproductive biology (when feasible) approaches for best practices in defining novel fungal species. Both MLST schemes we applied showed a clear differentiation of the *Parengyodontium* cluster among other well-characterized lineages clustered within the Cordycipitaceae (Figure 3, Appendix A). The *Parengyodontium* cluster was supported by both bootstrap and aLRT values (90.7/78) and was closely related to the *Simplicillium* species, *Lecanicillium aranearum,* and *Lecanicillium antillanum*. As *L. kalimantanense* BTCC-F23, and *T. wallacei* CBS 101237 also clustered within the *Parengyodontium* lineage, a careful examination of these two particular strains utilizing polyphasic taxonomy by including different isolates of the related genera (*Lecanicillium* and *Torrubiella*) is needed. In parallel to a single gene or multi-genes phylogeny, technological improvements to genome sequencing offered promising alternatives [81]. Using whole-genome phylogenetic analysis that relied on 5334 single-copy orthologs protein alignments, we also showed that *Parengyodontium* forms a monophyletic cluster among other Cordycepatecae lineages and *P. torokii* is highly differentiated from *P. americanum* (Figure 4). Molecular systematics analysis based on WGS of the misidentified strains might help to better understand its phylogenetic position along with other members of the *Parengyodontium* complex. In this communication, using microscopy, MLST analyses, and WGS, the phylogenetic novelty of *P. torokii* strains was defined.

Although not a major concern for robotic missions, long duration crewed missions may give microorganisms adequate time to form biofilms within vulnerable systems, putting both crew health and spacecraft longevity in jeopardy. The development of nanoengineered materials that prevent and mitigate biofilm formation is significant to current and future NASA missions. Despite reports of environmental and clinical biofilms containing mixtures of bacteria and fungi [82,83], most biofilm studies focus on classic bacterial biofilm formers (e.g., *Pseudomonas aeruginosa*, *Staphylococcus aureus*) [84]. The results of this study demonstrated that Teflon and the polypropylene both acted as a substrate for substantial biofilm formation by the *P. torokii* FJII-L10-SW-P1 strain. Indeed, even the addition of antibiofilm coatings could not prevent *P. torokii* isolate from colonizing Teflon and producing a robust biofilm of comparable size and thickness to the untreated surface (Figure 2). In this study, we showed that with both coated and uncoated Teflon surfaces there is a minimal amount of biofilm mycelium attachment at the surface of the coupon substrate, leading to the formation of a canopy morphology on both uncoated and coated coupon substrates. This minimal amount of mycelium between the coupon and the upper canopy of the biofilm could be due to extracellular polymeric substances (EPS) or other secreted compounds attaching the fungal biomass to the substrate (Appendix A). The confocal stains used were targeting nucleic acids (ToPro-3) and cellulose (Calcofluor-white); thus, any secreted EPS compounds lacking nucleic acid or cellulose would not have appeared in these images. It is unclear if material is present, and if so, whether it was secreted after attachment to the surface, or whether the initial colonizing mycelium/fungal conidia had a thick layer of a coating (Appendix A). If the colonizing mycelium/fungal conidia had such an initial coating, this could impart some protection from contact with the antimicrobial coating. SEM analysis of vegetative cells of *P. torokii* FJII-L10-SW-P1 strain (Figure 1E,F and Appendix A) show the presence of a thin membranous layer that may be linked to this phenomenon. 

Furthermore, we identified a wide number of putative orthologues in *P. torokii* involved in the production of complex polysaccharides as well in the biofilm formation and adhesion that were well characterized in other model fungal organisms (Appendix A) [85]. Confocal analysis demonstrated that the biofilm was smaller in size and shorter in height than the biofilm formed on the coated surface. It is not clear if the biofilm formed on the coated surface was further away due to the presence of the antimicrobial coating, or if the larger size of the biofilm forced the biofilm higher. What is clear is that the novel filamentous fungal isolate described here is a strong biofilm former; additionally, this study highlights the importance of testing antimicrobial surfaces with microbes other than the classic bacterial biofilm formers, *Pseudomonas aeruginosa* and *Staphylococcus aureus*. The antimicrobial compound used in this study was selected because it demonstrated substantial biocidal activity against both of these classic bacterial biofilm formers [25]; however, our study here indicates that when filamentous fungus is grown in the presence of the coating, the biofilm is actually larger (Figure 2). The reason for this enlarged biofilm on the coated surfaces as compared to the uncoated surfaces is not known, and further testing with microorganisms under different growth conditions could help to explain this finding. Environmental biofilms are usually multi-species [86] and can contain a mixture of bacteria and fungi [83,87]; despite this reality, very few biocidal coatings are effectively tested against fungus or mixed communities of microbes before coming on to the market.

The detailed comparison of the genomic annotation of *P. torokii* to closely related *Parengyodontium* species and other Cordycipitaceae fungi identified some enzymes families that are uniquely expanded in the *Parengyodontium* sp. For example, the CAZy Glycosyl Hydrolase GH33 family is unique for the *Parengyodontium* group of fungi not shared with other Cordycipitaceae species. This enzyme, also known as sialidases (E.C. 3.2.1.18), is responsible for hydrolyzing a wide variety of N-acetylneuraminic acids linked to various types of sugars. Such chemical modifications can alter the structure of glycoconjugates to which they are linked, thus modifying intermolecular and intercellular interactions with hosts and pathogens or symbionts, for example [88]. We also observed an increase of the CAZy Glycosyltransferases families GT20 and GT34 compared to its related species. Glycosyltransferases play a pivotal role on the biosynthesis of disaccharides, oligosaccharides, and polysaccharides by transferring glycosyl moieties, forming glycosidic bonds, from activated donor molecules to other sugar molecules [89]. More specifically, members of the GT20 family (2.4.1.15) participate in the trehalose biosynthesis and GT34 members in galactomannan production. These pathways are involved in biofilm formation and development, cell morphogenesis, cell wall maintenance, metabolism dynamics, and virulence [90].

*Parengyodontium* species produce compounds that are biotechnologically valuable, such as various proteases [91], leading to the commercial production of proteinase K. We found two putative orthologues of the proteinase K gene in the FJII-L10-SW-P1 genome with high similarity to the original sequence described for *P. album* (Appendix A) [92]. The WGS analysis of the LEC01 strain, whose genome is nearly identical to the Mars 2020 strain (FJII-L10-SW-P1), produced chitinase, chitosanase, cutinase, hydrolase, and lipase [7]. Proteases are capable of breaking down protein substrates using a universal mechanism of activation of water by Zn^2+^ ions and display multiple physiological functions [93]. Different families of metallopeptidases produced by pathogenic fungi are related to critical biological processes as well pathogenesis to human, plants, and insects [77]. More specifically, enzymes of the M24B family cleavage the N-terminal amino acid from peptides harboring a proline residue at the second position and are essential for cellular metabolic processes [94]. The members of the M28E family are characterized as aminopeptidases that are important for protein turnover and virulence [95,96]. Lastly, the function of metalloprotease belonging to the family M35 has been widely investigated in fungal genomes [76]. These proteins harbor two zinc-binding histidines and a glutamate catalytic residue located in a HEXXH motif, and they are involved in a series of pathogenic processes of different pathogenic fungi [76]. More research is needed to confirm and verify the pathogenicity of *P. torokii*.

Based on the WGS analyses, we characterized 32 secondary metabolite clusters predicted by antiSMASH (Appendix A), suggesting that this fungus may produce a wide range of bioactive compounds. These small molecules are extremely important for microbe-host and microbe-microbe interactions and might have biotechnological interest. One metabolite predicted in silico was also found using the LC-MS approach, specifically cytochalasin K, and is putatively [78] biosynthesized by this fungus. This biosynthetic cluster was also conserved among other Pezizomycotina fungi (Figure 8). Several in vitro and in vivo studies were conducted on the anticancer activity of cytochalasins [97], as the compound was found to influence the end stages of mitosis and elicited a profound synergistic effect on cancerous cells [98]. Furthermore, the secondary metabolite clusters predicted from the genome, such as equisetin, cephalosporin C, EQ-4, squalestatin S1, curvupallide-B, pyranonigrin E, and dimethylcoprogen, were not found to use these LC-MS approaches [19,20], and revealed that further studies are warranted to potentially confirm their biosynthesis by *P. torokii.* It is also possible that these compounds might have been produced at a quantity not detected by the LC-MS employed in this study or are expressed under specific conditions of growth. 

## 6. Conclusions

In summary, a novel fungal species was isolated from the Mars 2020 spacecraft assembly facility and identified as *P. torokii* using morphological characteristics and multi-gene phylogenetic analyses. Biofilm formation by this fungus could be problematic and suitable countermeasures should be developed to preemptively inhibit the growth and persistence of this microorganism in cleanroom facilities, where sensitive space instrumentations are assembled for both robotic and crewed missions. The comparative genome analysis and subsequent genome annotation revealed the presence of gene clusters responsible for biofilm production. One fungal metabolite, cytochalasin K, was predicted in the WGS analysis and confirmed using the LC-MS metabolic analysis.

## Figures and Tables

**Figure 1 jof-08-00066-f001:**
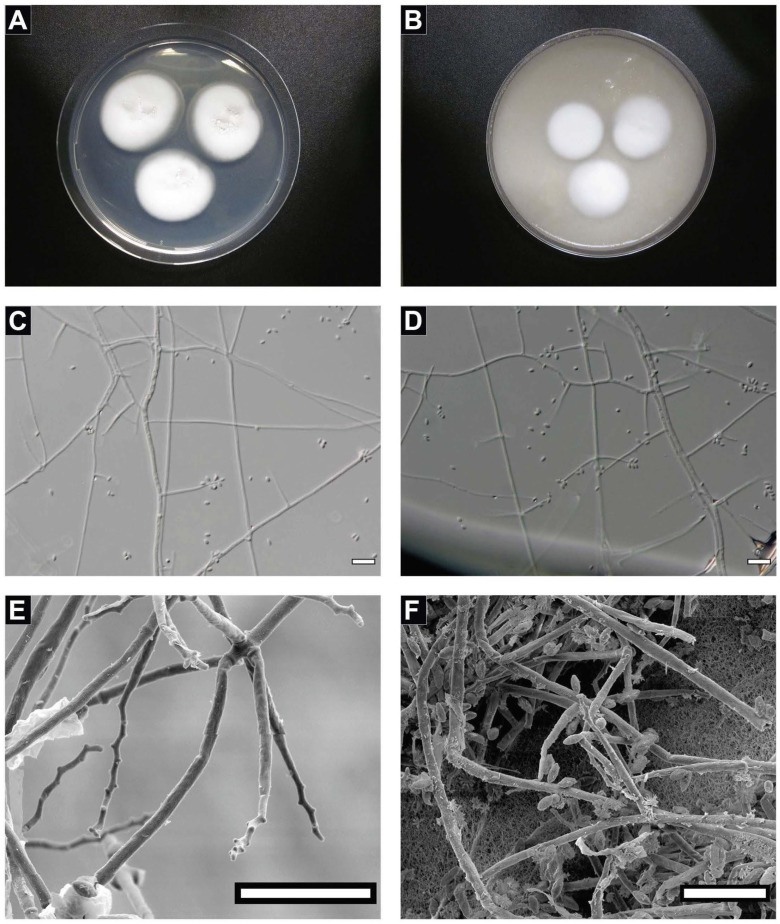
**Macro and micromorphology of *Parengyodontium torokii*.** Colony surface of FJ11-L10-SW-P1 after 21 days of incubation at room temperature (23 °C) in standard 9 cm petri dishes on (**A**) PDA media and (**B**) OMA media. (**C**) Conidia produced at each bent point of the zigzag rachides of the fertile conidiogenous cells. (**D**) Whorl of two conidiogenus cells with conidia attached at the zigzag rachides. Scale Bars (**C**–**E**) = 20 µm. (**E**) Scanning electron microscopy images of *Parengyodontium torokii* from ex-type strain FJ11-L10-SW-P1 with whorl of conidiogenous cells showing butt-shaped denticles and (**F**) subcylindrical to ellipsoidal, hyaline single-celled conidia. Scale bar for all microscopy is 10 µm.

**Figure 2 jof-08-00066-f002:**
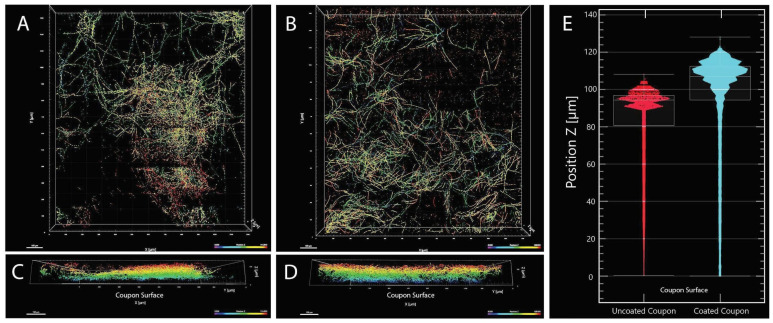
**Confocal analysis of biofilm formation.***Parengyodontium torokii* isolate FJII-L10-SW-P1 was grown in PDB in the presence of untreated (**A**,**C**) and treated (**B**,**D**) Teflon coupons. Each scan was 1161 µm × 1336 µm and was taken at a representative location on the coupon surface. Composite confocal fungal space-filling structure is color-coded based on relative distance from the coupon surface with blue/violet being proximal to the surface and orange/red being distal. (**A**,**B**) are plan views of the region that was imaged (100 µm scale bars) while (**C**,**D**) are orthogonal views of the biofilm looking length-wise through the biomass (100 µm scale bars). The comparative biofilm distribution as compared to distance from the coupon surface is presented in (**E**). The biofilm formed on the uncoated Teflon is smaller and the median height is closer to the surface while the biofilm from the coated Teflon coupon is larger and has a higher median height (94 µm and 107 µm, respectively).

**Figure 3 jof-08-00066-f003:**
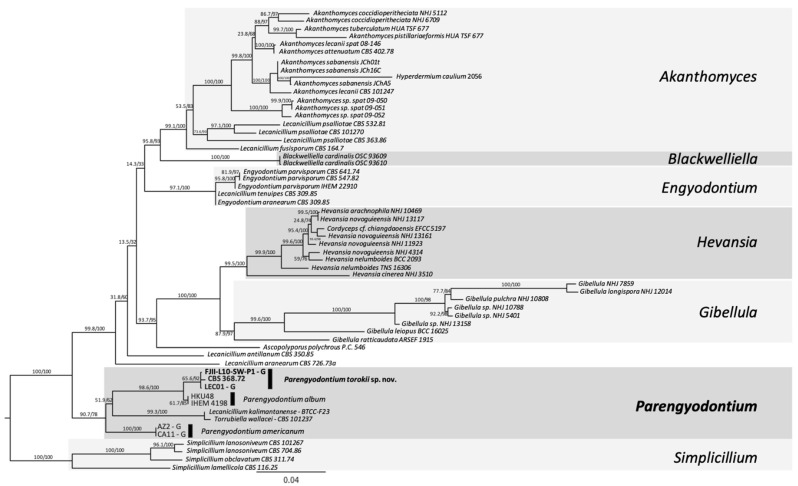
**Multi Locus Sequence Typing (MLST) of *Parengyodontium torokii*.** Gene sequences from the ITS region rRNA gene, D1/D2 domain of large subunit (LSU) rRNA gene, small subunit (SSU) rRNA gene, RNA polymerase II (*RPB1* and *RPB2*), and translation elongation factor 1-α (*TEF1*) were used to investigate phylogenetic placement of the FJII-L10-SW-P1 among the main Cordycipitaceae groups. We used 59 taxa and 4617 nucleotide sites to build up a Maximum Likelihood tree on the IQTREE2 software. The branches are proportional to the number of mutations and 1000 ultrafast bootstraps and SH-like approximate likelihood ratio test (aLRT) was used to test branch support and added to each corresponding branch of the tree. The tree was rooted with the *Simplicillium* sp.

**Figure 4 jof-08-00066-f004:**
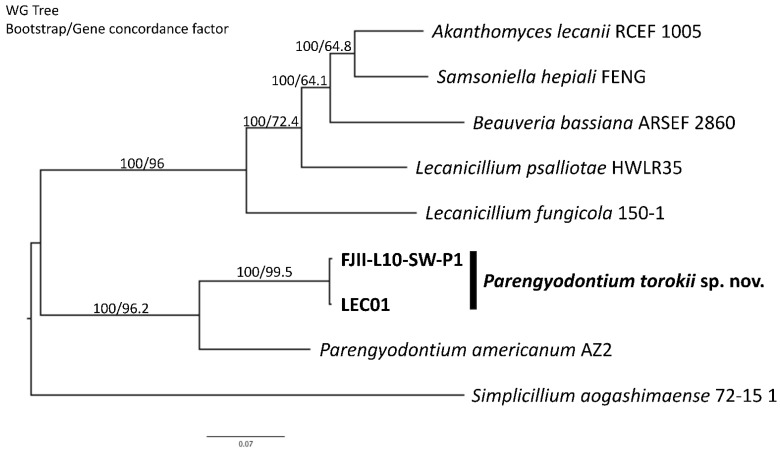
**Phylogenomic analyses of *Parengyodontium torokii*.** A total of 5334 single copy orthologous genes were used to build up a Maximum Likelihood tree among 9 Cordycipitaceae fungi using the IQTREE2 software. *Simplicillium aogashimaense* was set as the outgroup and the branches are proportional to the number of mutations. Branch fidelity used two different approaches, Ultrafast bootstraps and Gene Concordance Factors, which were added next to its corresponding branches.

**Figure 5 jof-08-00066-f005:**
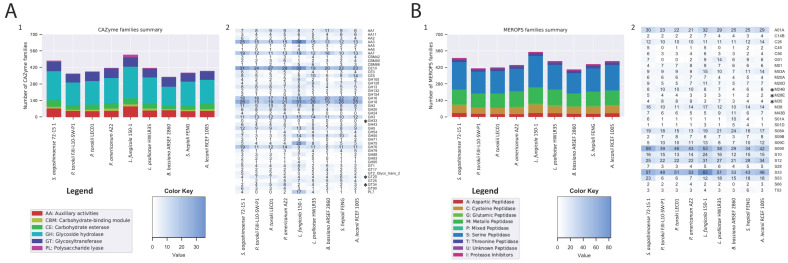
**Comparative genomic analyses of Cordycipitaceae fungi.** (**A**) Carbohydrate-active enzymes (CAZymes) and (**B**) proteases (MEROPS). Each category with a standard deviation > 1 was plotted in the heat map. Asterisks represent the more prominent changes in *Parengyodontium lineage* in both CAZy and MEROPS classes. The numerical number “1” represents for the major families shown in bar plot and “2” depicts the significantly enriched or depleted functional traits.

**Figure 6 jof-08-00066-f006:**
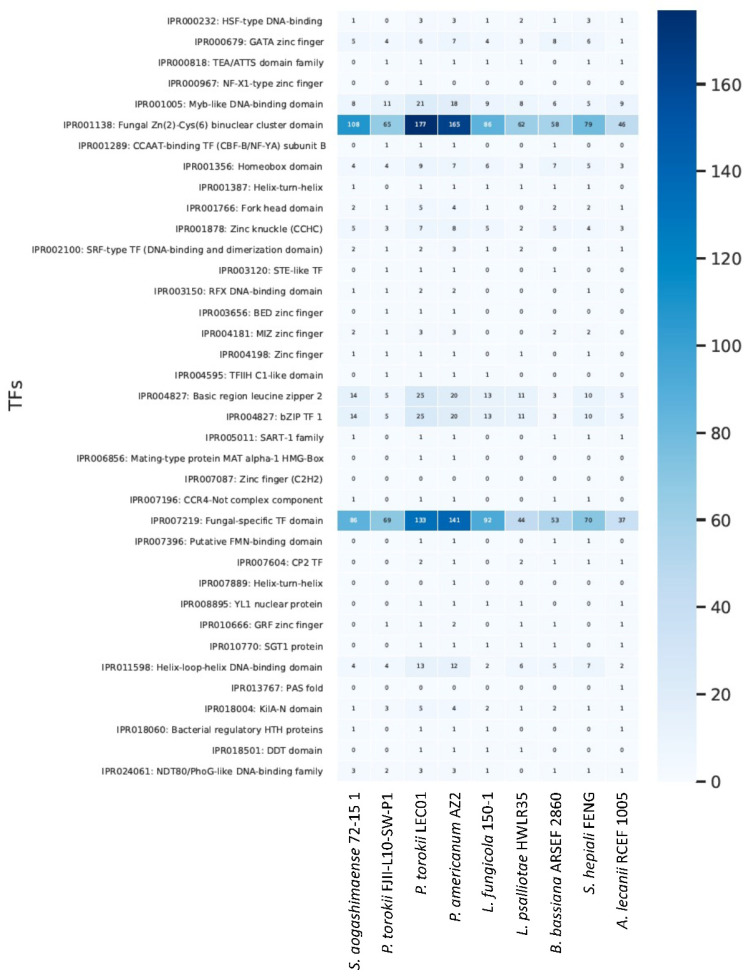
**Fungal-specific transcriptional factors among the Cordycipitaceae species.** Fungal transcription factors were screened in 9 Cordycipitaceae genomes and compared. Factors with a standard deviation >1 was plotted in a heat map.

**Figure 7 jof-08-00066-f007:**
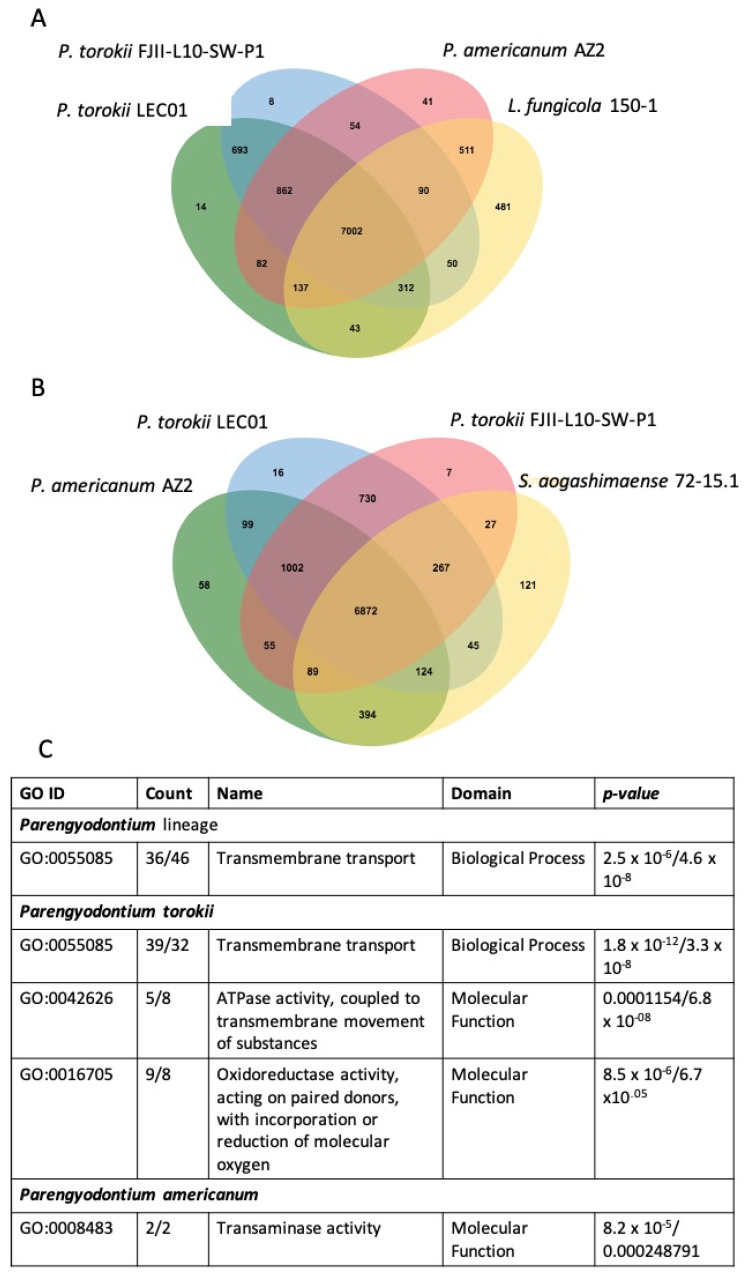
**GO-enrichment of the *Parengyodontium* lineage.** The Venn diagram shows shared orthologs and unique groups of genes of the *Parengyodontium* species using either *Lecanicillium fungicola* 150-1 (**A**) or *Simplicillium aogashimaense* 72-15.1 (**B**) as outer species for comparisons. (**C**) The GO categories shared in both scenarios enriched for either *Parengyodontium* sp., *Parengyodontium torokii*, or *Parengyodontium americanum* are listed with its respective function, domain, number of counts, and associated *p*-value.

**Figure 8 jof-08-00066-f008:**
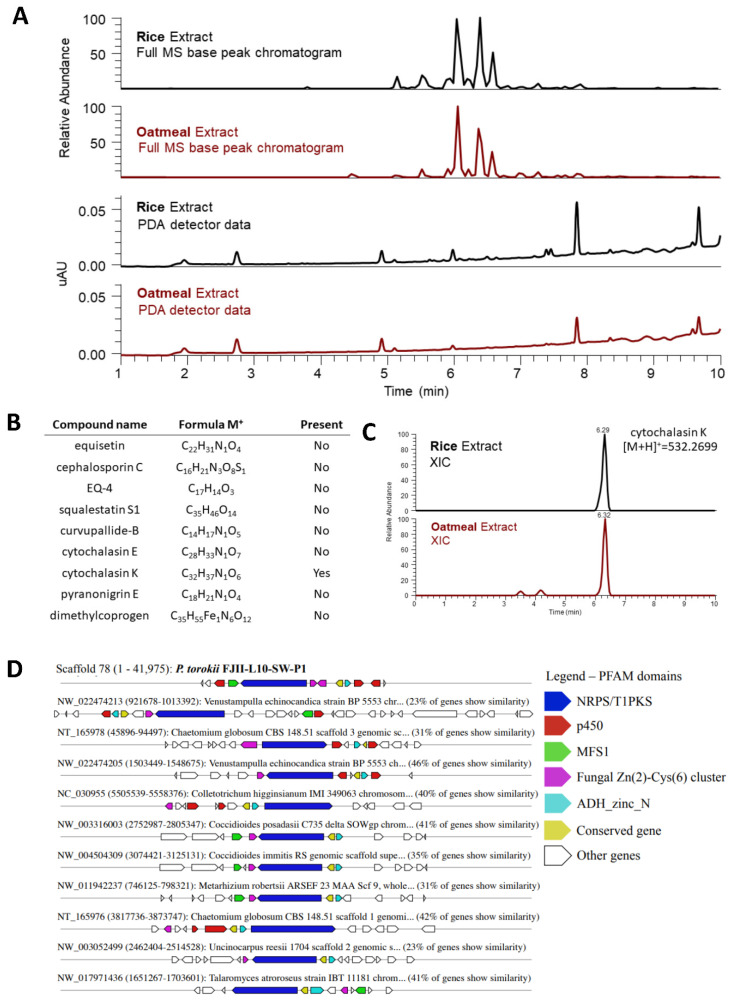
**LC-MS comparison of FJ11-L10-SW-P1 grown on oatmeal and rice media.** (**A**) The fungus was grown on two different media to examine its secondary metabolite production. Extracts of the fungus were examined through high-resolution mass spectrometry (HRMS). The base peak chromatogram along with light absorbance data collected through Photodiode-Array Detection (PDA) was compared. The extracts showed high similarities in their metabolomic profiling. (**B**) The fungal extracts were screened for natural compounds that are known to be expressed by the identified biosynthetic gene clusters. (**C**) The accurate mass of cytochalasins K was identified through extracted-ion chromatogram (XIC) within 5 ppm of the compound’s accurate mass value. Thus, cytochalasin K was putatively biosynthesized by this fungus under the used growth conditions. (**D**) Predicted cluster of cytochalasin K identified in the *P. torokii* FJ11-L10-SW-P1 genome. The scaffold and positions spanning the biosynthetic cluster as well the genes are displayed along the schematic representation. The same cluster was identified for a series of other filamentous fungi, which are also shown.

**Table 1 jof-08-00066-t001:** Summary of the draft whole-genome sequences of *Parengyodontium* and closely related species belonging to the family Cordycipitaceae.

	*Parengyodontium toroki* FJII-L10-SW-P1	*Parengyodontium toroki* LEC01	*Parengyodontium americanum* AZ2	*Akanthomyces lecanii* RCEF 1005	*Simplicillium aogashimaense* 72-15 1	*Lecanicillium fungicola* 150-1	*Lecanicillium psalliotae* HWLR35	*Beauveria bassiana* ARSEF 2860	*Samsoniella hepiali* FENG
**Assembly**									
# of contigs	440	352	295	131	20	782	197	239	222
Genome size	30,424,506	31,084,693	32,962,623	35,580,375	29,244,117	44,547,425	36,133,949	33,693,821	34,650,604
Largest contig	718,708	1,200,404	821,300	5,461,016	4,930,463	493,648	4,365,396	2,084,429	1,229,925
Repetitive DNA (%)	2.83%	7.31%	9.98%	11.29%	7.38%	8.44%	11.34%	11.78%	11.91%
GC (%)	50.45	50.28	52.88	53.10	49.01	49.87	52.73	51.36	53.89
N50	122,374	310,369	345,622	3,613,853	3,162,613	154,124	2,330,369	724,305	576,310
L50	70	27	31	4	4	92	6	13	20
**Annotation**									
tRNA	70	77	95	115	85	144	121	111	115
intron	16,303	16,562	16,709	14,195	15,835	19,694	13,365	15,912	13,429
Exons	25,899	26,363	27,008	24,306	25,962	32,964	23,566	25,710	23,649
average exon length	478	482	472	494	483	476	502	476	518
mRNA	9596	9801	10,299	10,111	10,127	13,270	10,201	9798	10,220
CDS	9596	9801	10,299	10,111	10,127	13,270	10,201	9798	10,220
gene	9666	9878	10,394	10,226	10,212	13,414	10,322	9909	10,335
average gene length	1658	1642	1570	1565	1579	1536	1521	1627	1560
average protein length	496	501	479	484	489	470	472	498	482
**Functional**									
go_terms	2913	5980	6243	1832	3109	2750	2306	2301	2947
interproscan	3915	8073	8454	2523	4184	3879	3206	3118	4061
eggnog	9330	9455	9892	9736	9780	12,457	9749	9461	9793
pfam	6961	7187	7476	7269	7530	9092	7160	7001	7141
cazyme	355	370	399	380	459	507	403	331	364
merops	412	418	438	472	498	549	471	402	447
busco	3685	3742	3739	3661	3748	3755	3596	3747	3573
secretion	826	899	995	1060	1144	1403	1144	1008	1009

## Data Availability

The draft genome (JADQAY000000000) and raw data (SRR12385174) have been deposited in GenBank under the BioProject accession number PRJNA644637.

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
