# Peer review of "Genomic Characterization of Parengyodontium torokii sp. nov., a Biofilm-Forming Fungus Isolated from Mars 2020 Assembly Facility"

_jof, 2022, doi:10.3390/jof8010066_

Round 1
Reviewer 1 Report
This study reported a new species, Parengyodontium torokii, which isolated from the Mars 2020 spacecraft assembly facility, and exhibited biofilm formation. The morphological characteristics and phylogenetic analysis were both robust and supports the conclusions. The biofilm formation was well supported by the genomic informtion.
But I would like to suggest the authors that the content needs to be fine tuned, and more highlight the title, the new species and its biofilm. There is a suggestion that the mitochondrial genome information of the strain was not important to the main text and could be delete. Besides, there are a few types and grammatical corrections necessary, and minor proofreading is required. Such as, line 534, "16 – 22 μm x 1 – 2 μm"; line 536, "2-3 μm x 1-2 μm" , the format needs to be unified.
Author Response
Rev 1:
This study reported a new species, Parengyodontium torokii, which isolated from the Mars 2020 spacecraft assembly facility, and exhibited biofilm formation. The morphological characteristics and phylogenetic analysis were both robust and supports the conclusions. The biofilm formation was well supported by the genomic informtion.
But I would like to suggest the authors that the content needs to be fine tuned, and more highlight the title, the new species and its biofilm.
Ans: More efforts were made by moving the taxonomy portion to the earlier part of the Results section, language checked, and biofilm discussion added.
There is a suggestion that the mitochondrial genome information of the strain was not important to the main text and could be delete.
Ans: The identification, assembly and annotation of the mitogenome is important to split nuclear from mitochondrial DNA content of a given organism and is part of the AAFTF pipeline. We moved the mitogenome map into the supplementary material.
Besides, there are a few types and grammatical corrections necessary, and minor proofreading is required. Such as, line 534, "16 – 22 μm x 1 – 2 μm"; line 536, "2-3 μm x 1-2 μm" , the format needs to be unified.
Ans: Modified.
Reviewer 2 Report
Review of “Genomic characterization of Parengyodontium torokii sp. nov., a biofilm-forming fungus isolated from Mars 2020 assembly facility”
General Comment
The manuscript addressed the characterization of a fungal isolate with a biofilm forming capability, found at the NASA’s Mars 2020 spacecraft assembly facility. The manuscript is well written, and presents interesting results. However, there are details that need to be clarified or rectified.
Major points
Overall, the work conducted for this manuscript was consisted of three parts: 1) taxonomic and phylogenetic studies, 2) characterization of biofilm and metabolites, and 3) phylogenomics and comparative genomics analyses.
- Taxonomic and phylogenetic studies
For the first part, the results clearly showed a new species, named as Parengyodontium torokii. The status of the new species was reliably demonstrated based on the phylogenies but the morphological characterization was relatively elusive. Some details need to be clarified.
1.1 For example, why conducting two separate MLST analyses? The first MLST analysis included ITS, LSU and B-Tub. The second included ITS, LSU, SSU, RPB1, RPB2 and Tef1. Why just not combining all the markers together and make a phylogeny based on 7 markers instead of doing two separated analyses with 3 and 6 loci? This was not very well explained (the section “MLST-based phylogenetic analysis”). It seemed that the reason behind this implementation is that the taxa included do not have equivalently the sequences of these loci (from my experience, only some fungi of Cordycipitaceae have Beta-tubulin data). If this is the case, it should be explicitly explained in M&M.
1.2 It is unclear how many strains exactly were obtained after isolation and included in the phylogenetic analyses. L218 – L221 indicate that 75 fungal strains were isolated, but only one was included in the phylogenetic analyses. In this case, all other isolates in the Figure 3, S3 and S4 should have a table showing sequences accession numbers and bibliographic references to the original studies for these strains. These information are not explicit enough in the manuscript.
If there are several strains isolated, but only one used in this study. This fact should be mentioned since the introduction.
1.3 The mention of P. album subclades 1, 2 and 3 since the abstract and the introduction makes it difficult to understand if the readers have not yet read through to the results. I’m not sure whether these subclades are already discovered in previous studies on the taxonomy of P. album. I suggest the authors to be clearer on this. There should also be a mention in the manuscript that the subclades 1 and 2 could be separated as distinct species.
1.4 The six-loci phylogeny (mentioned as five-gene phylogeny in the manuscript) does not include some important genera such as Cordyceps, Beauveria and Samsoniella. At least, maybe Beauveria should be included as it is the genus in which Parengyodontium was once mis-classified into. It was not very clear why the 58 fungal strains were used…because they all had all the six markers available? As raised in the point 1.2 above, without a table with sequence accessions, it is unclear whether the sequences of the markers are well covered for the study or not. This might result in some artefacts.
1.5 L732 – L734: The branching of the clade of L. kalimantanense and T. wallacei into the Parengyodontium was not strongly supported. It is highly possible that the branching there is an artefact from the sequences or due to too many loci lacking for these two fungi. Please check single gene phylogenies (to be shown in supplementary materials) to find where the incongruence lies, and identify potential errors in the alignments, and therefore propose a better discussion on this issue.
1.6 The demonstration of the morphological difference between P. torokii and the closely related P. album and P. americanum was elusive. L413 mentions that the strain FJII-L10-SW-P1 does not share morphological characteristics of spores with P. album, but then at the end of the paragraph it is said that the SEM of CBS 368.72 (P. album) exibited similar micro and macromorphologies… So are these two species different or not? If there is some difference, it is great. But if it is not, then the notion of cryptic species can be evoked.
1.6.1 There should be a table making a morphological comparison between new species proposed with closely related species. This is to make clear whether there is difference or not.
1.6.2 Some P. album strains are deposited at CBS. They could be studied for their colony characteristics and micromorphologies with statistical analyses to compare with P. torokii. This is just a suggestion.
- Characterization of biofilm and metabolites
I have no critique on this part of the manuscript. From what I read, it is comprehensible and interesting. The fact that the authors showed that even coated materials cannot avoid the development of biofilms from the studied fungus was interesting. Some clarifications are needed though.
2.1 How many coupons were used per material type (Inconel vs. Teflon) and per treatment (coated vs. uncoated).
2.2 The Figure 2 presented the results of confocal microscopy analysis on Teflon coupon. What about Inconel? Is there any difference between the two types of materials.
2.3 Just one clarification needed. L377: what does the “two media” referred to? PDA and liquid PDA? PDA and solid fermentation media? Or the solid fermentation media with rice or oatmeals. It’s not very clear here. Please reformulate this passage.
- Phylogenomics and comparative genomics analyses
Overall, this part was well executed. The pipelines used for the genome annotations were appropriate. However, some conclusions were overhasty and clarifications are needed.
3.1 Is is not clear why the other 8 fungal genome assemblies were selected for use this study? For example, for B. bassiana, there are other genome assemblies which is better in representation (few big scaffolds approaching chromosomes). Cordyceps militaris, which is a very well known species with many known metabolites, was not included while it may be interesting to compare the secondary metabolite gene clusters between them. It is ok to pick random representative from different genera within Cordicipitaceae, but this should be said.
3.2 L581 – L582 gave some precipitate conclusions regarding the number of genes (lower in Parengyodontium) and the gene length (longer in Parengyodontium). The tendency observed by the authors was not so significant and there is no statistical support. I suggest tempering their statements in this part.
3.3 L624: the authors stated that there was an enrichment of transcriptional factors. This statement is inaccurate. From what I see from the Fig.6D, only a few tRNA families are enriched in Parengyodontium... The sentence should be changed.
3.4 Figure 6: The heatmaps on this figure are not very visible, the labels for the rows and the numbers in the tables are unreadable (particularly for 6D). Improve the figure or rather put Fig.6D in the supplementary materials.
3.5 L655 – L656: it is unclear why the authors used these two species (Simplicillium aogashimaense and Lecanicilium fungicola) as outgroup for GO terms analysis.
GO terms analysis can be done among Parengyodontium species to identify species-specific genes and their functional enrichment. In general, the comparison between species is done if there are species-specific functions expected, or between set of species that have different ecology or life styles as we might expect enrichments related to specific functions. If the authors want to identify GO categories enrichment specific to the genus Parengyodontium (as announced in the Materials and Methods: L363 – L365), then genes specific to this genus must be identified relative to the whole set of genes from all the other genomes included. How these analyses were done and presented did not make much sense to me.
Minor points
Beside the points raised above, I have some minor comments below.
- L31: the mention “T” as superscript to the strain code is not necessary and can lend to confusion.
- L57: “Parengyodontium album subclade 3”…confusing as it is mentioned here (see my comment 1.3 above).
- L71 – L73: “The ITS-based phylogenetic analysis and MALDI-TOF profiles of several P. album strains showed three distinct subclades, whereas the 28S rDNA-based phylogeny could not separate subclades 1 and 2.” Is this sentence referred to another study? If so, please give a citation. If not, please change the way you present this as I commented above (comment 1.3).
- L264 and L267: should be six loci? (ITS, LSU, SSU, RPB1, RPB2, Tef1).
- L302: give a reference for the funannotate pipeline.
- Table 1: the proportion of repeats in the genomes should be given in this table.
- L357: the name of the softwared used to doing the Ultrafast bootstraps and gene concordance factor should be given here.
- L363: p-value < 0.05…based on what statistical analysis?
- L379: how much was the volume of inoculum from liquid PDA to the solid state fermentation media? Was there an adjustment for the concentration of the fungus in the inoculum? Please clarify.
- L429: sensu stricto, not the reverse please.
- L447: the uncoated and antimicrobial coated coupons
- L485 and L491: “the six-loci…”, rather than “the five-gene…”
- L675: “to verify similarities”, rather than “to find…”
- L843: I suggest using the terms “The comparative genomics analyses revealed…”
- Table S1: please give the references associated to the genomes used in this study if there are any.
Author Response
Rev new (1):
General Comment
The manuscript addressed the characterization of a fungal isolate with a biofilm forming capability, found at the NASA’s Mars 2020 spacecraft assembly facility. The manuscript is well written, and presents interesting results. However, there are details that need to be clarified or rectified.
Major points
Overall, the work conducted for this manuscript was consisted of three parts: 1) taxonomic and phylogenetic studies, 2) characterization of biofilm and metabolites, and 3) phylogenomics and comparative genomics analyses.
- Taxonomic and phylogenetic studies
For the first part, the results clearly showed a new species, named as Parengyodontium torokii. The status of the new species was reliably demonstrated based on the phylogenies but the morphological characterization was relatively elusive. Some details need to be clarified.
Ans: The taxonomy part is revamped and details are given in the modified manuscript.
1.1 For example, why conducting two separate MLST analyses? The first MLST analysis included ITS, LSU and B-Tub. The second included ITS, LSU, SSU, RPB1, RPB2 and Tef1. Why just not combining all the markers together and make a phylogeny based on 7 markers instead of doing two separated analyses with 3 and 6 loci? This was not very well explained (the section “MLST-based phylogenetic analysis”). It seemed that the reason behind this implementation is that the taxa included do not have equivalently the sequences of these loci (from my experience, only some fungi of Cordycipitaceae have Beta-tubulin data). If this is the case, it should be explicitly explained in M&M.
Ans: The reason for using two datasets is exactly the number of taxa/markers overlapping the two MLST datasets. Indeed the 6-gene MLST dataset is the most complete one and recurrently used to diagnose Cordycipitaceae species. We appreciated the comments and this information was added to the M&M section (Lines 245-247).
1.2 It is unclear how many strains exactly were obtained after isolation and included in the phylogenetic analyses. L218 – L221 indicate that 75 fungal strains were isolated, but only one was included in the phylogenetic analyses. In this case, all other isolates in the Figure 3, S3 and S4 should have a table showing sequences accession numbers and bibliographic references to the original studies for these strains. These information are not explicit enough in the manuscript.
If there are several strains isolated, but only one used in this study. This fact should be mentioned since the introduction.
Ans: The following details were placed in Line # 225 to 230.
- Among the 75 fungal isolates, 12 strains were novel based on ITS-sequence analyses (data not shown); however, we have performed detailed phylogenetic analyses only for the FJII-L10-SW-P1 strain due to its capability in forming biofilm on spacecraft qualified materials. DNA was extracted from all fungal strains (n=75 isolates) using the Maxwell-16 MDx automated system following the manufacturer’s instructions (Promega, Madison, WI). Initial identification of the fungus was performed by amplicon sequencing targeting the internal transcribed spacers (ITS) region.
1.3 The mention of P. album subclades 1, 2 and 3 since the abstract and the introduction makes it difficult to understand if the readers have not yet read through to the results. I’m not sure whether these subclades are already discovered in previous studies on the taxonomy of P. album. I suggest the authors to be clearer on this. There should also be a mention in the manuscript that the subclades 1 and 2 could be separated as distinct species.
Ans: The P. album subclades were discovered by Tsang et al., 2016. This was mentioned in Introduction and reproduced below with highlights. We also mentioned in this highlighted section that the cryptic species associated with subclades 1 and 2 need more study.
- The taxonomy of Parengyodontium is complex since its members were originally assigned to the genus Beauveria (Vuillemin, 1912), then to Tritirachium (Limber, 1940), and as Engyodotium (Gams et al., 1984). Finally, phylogenetic analyses targeting the ITS region, 28S nuclear ribosomal DNA, and β-tubulin gene as well as matrix-assisted laser desorption ionization–time of flight mass spectrometry (MALDI-TOF-MS) profiles, resulted in transferring members of the Engyodotium species to a novel genus, Parengyodontium, within the family Cordycipitaceae (Tsang et al., 2016). At the time of writing, the genus Parengyodontium consisted of album (Tsang et al., 2016) and P. americanum (Teixeira et al., 2020). The ITS-based phylogenetic analysis and MALDI-TOF profiles of several P. album strains showed three distinct subclades, whereas the 28S rDNA-based phylogeny could not separate subclades 1 and 2 (Tsang et al., 2016). The cryptic species associated with subclades 1 and 2 need more study, but during this study, the WGS-based phylogeny and multi-locus sequence type (MLST) analyses revealed that strains belonging to subclade 3 should be classified as a novel species of the genus Parengyodontium.
1.4 The six-loci phylogeny (mentioned as five-gene phylogeny in the manuscript) does not include some important genera such as Cordyceps, Beauveria and Samsoniella. At least, maybe Beauveria should be included as it is the genus in which Parengyodontium was once mis-classified into. It was not very clear why the 58 fungal strains were used…because they all had all the six markers available? As raised in the point 1.2 above, without a table with sequence accessions, it is unclear whether the sequences of the markers are well covered for the study or not. This might result in some artefacts.
Ans: We did not include the Cordyceps, Beauveria and Samsoniella genus because those are phylogenetically distant from the Parengyodontium/Engyodontium clades and we decided to include only Akanthomyces or other early-diverging Cordycipitaceae fungi. Moreover, the tree topologies from Wang et al., 2020 (https://doi.org/10.1007/s13225-020-00457-3), Kepler et al., 2016 (https://doi.org/10.5598/imafungus.2017.08.02.08) and Teixeira et al., 2020 (https://doi.org/10.1016/j.fgb.2020.10335) are concordant with our tree topology and thus we do not expect any conflicts/artefacts within the Parengyodontium clade and its position on the tree. The reason for using 58 taxa because those cover the diversity of the sister clades of the Parengyodontium lineage. We changed from five-gene to six-loci in the modified manuscript.
1.5 L732 – L734: The branching of the clade of L. kalimantanense and T. wallacei into the Parengyodontium was not strongly supported. It is highly possible that the branching there is an artefact from the sequences or due to too many loci lacking for these two fungi. Please check single gene phylogenies (to be shown in supplementary materials) to find where the incongruence lies, and identify potential errors in the alignments, and therefore propose a better discussion on this issue.
Ans: This is an interesting question but a bit complicated to draw any conclusions since we only have the full MLST scheme for the Torrubiella wallacei CBS101237. We only found the ITS marker for Lecanicillium kalimantanense at NCBI. To test if this close relationships of T wallacei/L. kalimantanense with the Parengyodontium clade, we performed a phylogenetic tree with 1.003 taxa targeting the ITS locus. We observed that three well defined clades current known for the Parengyodontium lineage. However, T. wallacei/L. kalimantanense and Parengyodontium are placed in different clades. Moreover L. kalimantanense is polyphyletic and the taxa appear in different clades along this tree. Further work is needed to better understand the phylogenetic relationships of these three fungi.
1.6 The demonstration of the morphological difference between P. torokii and the closely related P. album and P. americanum was elusive. L413 mentions that the strain FJII-L10-SW-P1 does not share morphological characteristics of spores with P. album, but then at the end of the paragraph it is said that the SEM of CBS 368.72 (P. album) exibited similar micro and macromorphologies… So are these two species different or not? If there is some difference, it is great. But if it is not, then the notion of cryptic species can be evoked.
Ans: Authors confirmed that members of subclade 3 including CBS 368.72 strain is indeed P. torokii. Hence the SEM of CBS 368.72 (Supplemental Figure S1) exhibited similar micro and macro-morphologies with the type strain of P. torokii. It is also clear form our study that the FJII-L10-SW-P1 strain does not share morphological characteristics of spores with P. album (subclade 1), but SEM of CBS 368.72 (subclade 3) exhibited similar micro and macro-morphologies of P. torokii type strain FJII-L10-SW-P1.
1.6.1 There should be a table making a morphological comparison between new species proposed with closely related species. This is to make clear whether there is difference or not.
Ans: Since we have only three species in Parengyodontium including P. torokii, we believe the text in the Taxonomy section is sufficient.
1.6.2 Some P. album strains are deposited at CBS. They could be studied for their colony characteristics and micromorphologies with statistical analyses to compare with P. torokii. This is just a suggestion.
Ans: We purchased CBS 368.72 and compared the morphologies (Suppl File Figure S1). Several attempts were made to procure other two strains of subclade 3 but did not succeed in getting them.
- Characterization of biofilm and metabolites
I have no critique on this part of the manuscript. From what I read, it is comprehensible and interesting. The fact that the authors showed that even coated materials cannot avoid the development of biofilms from the studied fungus was interesting. Some clarifications are needed though.
2.1 How many coupons were used per material type (Inconel vs. Teflon) and per treatment (coated vs. uncoated).
Ans: All experiments were carried out in triplicate but SEM studies were conducted from single coupon.
2.2 The Figure 2 presented the results of confocal microscopy analysis on Teflon coupon. What about Inconel? Is there any difference between the two types of materials.
Ans: The following information was present in the manuscript and reproduced below.
- Biofilms were not formed on either the glass-bottomed Petri dishes or on the Inconel coupons. Both borosilicate glass surfaces and Inconel (a nickel-chrome superalloy) are smooth surfaces, while Inconel is additionally resistant to corrosion and oxidation.
2.3 Just one clarification needed. L377: what does the “two media” referred to? PDA and liquid PDA? PDA and solid fermentation media? Or the solid fermentation media with rice or oatmeals. It’s not very clear here. Please reformulate this passage.
Ans: They are just rice and water in one medium and oatmeal and water in another medium.
- To prepare the two media, 10 g of rice and oatmeal were added to separate flasks (two rice and two oatmeal flasks) with 50 mL of deionized water.
- Phylogenomics and comparative genomics analyses
Overall, this part was well executed. The pipelines used for the genome annotations were appropriate. However, some conclusions were overhasty and clarifications are needed.
3.1 It is not clear why the other 8 fungal genome assemblies were selected for use this study? For example, for B. bassiana, there are other genome assemblies which is better in representation (few big scaffolds approaching chromosomes). Cordyceps militaris, which is a very well known species with many known metabolites, was not included while it may be interesting to compare the secondary metabolite gene clusters between them. It is ok to pick random representative from different genera within Cordicipitaceae, but this should be said.
Ans: We included genomes that represents the closest taxa of Parengyodontium among the Cordicipitaceae fungi. The reason for B. bassiana ARSEF 2860 is because this is the representative genome at NCBI, but indeed there are other genomes that are more complete in terms of completeness. Indeed, the genomes of B. bassiana and C. militaris are well annotated and were used to predicting the gene model of our new taxa. Regarding the secondary metabolite gene clusters, those are available at the Antismash pipeline and were used for comparative analysis. See Supplemental table S6 - full table
3.2 L581 – L582 gave some precipitate conclusions regarding the number of genes (lower in Parengyodontium) and the gene length (longer in Parengyodontium). The tendency observed by the authors was not so significant and there is no statistical support. I suggest tempering their statements in this part.
Ans: These statements were done based on the observations of the funannotate pipeline and no statistical analysis were performed. We made appropriated changes text. Please see lines 520-523
3.3 L624: the authors stated that there was an enrichment of transcriptional factors. This statement is inaccurate. From what I see from the Fig.6D, only a few tRNA families are enriched in Parengyodontium... The sentence should be changed.
Ans: Indeed, we described only families that are enriched for this particular lineage. Please see the Lines 551 to 556.
3.4 Figure 6: The heatmaps on this figure are not very visible, the labels for the rows and the numbers in the tables are unreadable (particularly for 6D). Improve the figure or rather put Fig.6D in the supplementary materials.
Ans: We split Figure 6 into 2 different figures and moved the Figure 6C to the supplement to make it more readable.
3.5 L655 – L656: it is unclear why the authors used these two species (Simplicillium aogashimaense and Lecanicilium fungicola) as outgroup for GO terms analysis.
GO terms analysis can be done among Parengyodontium species to identify species-specific genes and their functional enrichment. In general, the comparison between species is done if there are species-specific functions expected, or between set of species that have different ecology or life styles as we might expect enrichments related to specific functions. If the authors want to identify GO categories enrichment specific to the genus Parengyodontium (as announced in the Materials and Methods: L363 – L365), then genes specific to this genus must be identified relative to the whole set of genes from all the other genomes included. How these analyses were done and presented did not make much sense to me.
Ans: We used two different groups for comparisons (or “outgroups” - Simplicillium aogashimaense and Lecanicilium fungicola) to identify enriched GO terms for this particular lineage. We used genomes representative taxa from each sister clades of Parengyodontium (See WG-Tree). These two lineages represent the two sister clades of Parengyodontium, thus we used two different scenarios to propose different ecology or life styles for this particular group. Moreover, we described the enriched GO terms that were concordant in the two different scenarios. One limitation of Orthovenn is that this pipeline only takes a maximum of 6 genomes.
Minor points
Beside the points raised above, I have some minor comments below.
Ans: All of these minor points were incorporated.
- L31: the mention “T” as superscript to the strain code is not necessary and can lend to confusion. Removed.
- L57: “Parengyodontium album subclade 3”…confusing as it is mentioned here (see my comment 1.3 above). We provided the answer above
- L71 – L73: “The ITS-based phylogenetic analysis and MALDI-TOF profiles of several P. album strains showed three distinct subclades, whereas the 28S rDNA-based phylogeny could not separate subclades 1 and 2.” Is this sentence referred to another study? If so, please give a citation. If not, please change the way you present this as I commented above (comment 1.3). Yes Tsang et al., 2016 citation is added.
- L264 and L267: should be six loci? (ITS, LSU, SSU, RPB1, RPB2, Tef1). Changed to six loci.
- L302: give a reference for the funannotate pipeline. The following reference is cited https://zenodo.org/record/2604804 and the doi - 10.5281/zenodo.2604804
- Table 1: the proportion of repeats in the genomes should be given in this table. Updated.
- L357: the name of the softwared used to doing the Ultrafast bootstraps and gene concordance factor should be given here. We used IQTREE2 – this was updated in the manuscript.
- L363: p-value < 0.05…based on what statistical analysis? This was based on an FDR GO-enrichment p-value < 0.05.
- L379: how much was the volume of inoculum from liquid PDA to the solid state fermentation media? Was there an adjustment for the concentration of the fungus in the inoculum? Please clarify. 10 mL of fungal cultures grown for 7 days were inoculated into the solid-state fermentation media. Since the 7-day old culture will consists of both mycelia and conidia, we were not able to measure OD or CFU. However, the 7-day old culture was thoroughly homogenized by vigorously mixing using vortex and 10 mL aliquots of the liquid medium was transferred into fermentation media.
- L429: sensu stricto, not the reverse please. OK.
- L447: the uncoated and antimicrobial coated coupons Changed.
- L485 and L491: “the six-loci…”, rather than “the five-gene…” Changed.
- L675: “to verify similarities”, rather than “to find…” Changed.
- L843: I suggest using the terms “The comparative genomics analyses revealed…” Changed.
- Table S1: please give the references associated to the genomes used in this study if there are any. This table was updated with the references associated with the genomic studies.
Gams, W., De Hoog, G.S., Samson, R.A., and Evans, H.C. (1984). The hyphomycete genus Engyodontium a link between Verticillium and Aphanocladium. Persoonia - Molecular Phylogeny and Evolution of Fungi 12, 135-147.
Limber, D.P. (1940). A New Form Genus of the Moniliaceae. Mycologia 32, 23-30.
Teixeira, M.M., Muszewska, A., Travis, J., Moreno, L.F., Ahmed, S., Roe, C., Mead, H., Steczkiewicz, K., Lemmer, D., De Hoog, S., Keim, P., Wiederhold, N., and Barker, B.M. (2020). Genomic characterization of Parengyodontium americanum sp. nov. Fungal Genet Biol 138, 103351.
Tsang, C.-C., Chan, J.F.W., Pong, W.-M., Chen, J.H.K., Ngan, A.H.Y., Cheung, M., Lai, C.K.C., Tsang, D.N.C., Lau, S.K.P., and Woo, P.C.Y. (2016). Cutaneous hyalohyphomycosis due to Parengyodontium album gen. et comb. nov. Medical Mycology 54, 699-713.
Vuillemin, P. (1912). Beauveria, nouveau genera de Verticilla-cees. Bull. Soc. Bot. Fr. 59, 7.
Reviewer 3 Report
I have read on the manuscript “Genomic characterization of Parengyodontium torokii sp. nov., a biofilm-forming fungus isolated from Mars 2020 assembly facility”, ID JoF-1493606.
The article reports the isolation of Mar fungus. Fungal identification was based on morphological and molecular data as well as genomic. Biofilm formation was investigated. Functional genome was annotated and metabolomics was descripted. However, before the manuscript might be considered for publication in Journal of Fungi, it requires significant improvements:
- The morphological characteristics should be re-write and proofread by taxonomist and mycologist. The different of morphological characteristics of Parengyodontium album should be described and compared which show in the table format. Parengyodontium torokii was place in the same group of subclade 3with Parengyodontium album CBS 368.72, UAMH 9836 and LEC01. It’s better to compare their morphology with a new species.
- For the Taxonomy section, “Note:” should be add for comparison of the morphology and molecular data of a new species and close-related species.
- The phylogenetic species delimitation, what is the genealogical concordance phylogenetic species recognition (GCPSR) criterion used in this study? Please describe and mention. Figure of phylogenetic tree should be replace with the high resolution.
- The discussion should re-write in depth to elaborate discussion.
- Please verify the whole manuscript carefully for a journal format and editorial issues, such as italic and language mistakes.
- Some corrections showed in the attached file.

Author Response
I have read on the manuscript “Genomic characterization of Parengyodontium torokii sp. nov., a biofilm-forming fungus isolated from Mars 2020 assembly facility”, ID JoF-1493606.
The article reports the isolation of Mar fungus. Fungal identification was based on morphological and molecular data as well as genomic. Biofilm formation was investigated. Functional genome was annotated and metabolomics was descripted. However, before the manuscript might be considered for publication in Journal of Fungi, it requires significant improvements:
- The morphological characteristics should be re-write and proofread by taxonomist and mycologist. The different of morphological characteristics of Parengyodontium album should be described and compared which show in the table format. Parengyodontium torokii was place in the same group of subclade 3with Parengyodontium album CBS 368.72, UAMH 9836 and LEC01. It’s better to compare their morphology with a new species.
Ans: We compared the P. torokii strain isolated by us with CBS 368.72 after purchasing the same. We also tried to procure UAMH 9836 and LEC01 but could not be purchased from the culture collection (UAMH 9836) or from the investigator (LEC01). The comparative analyses of the type strain and CBS 368.72 isolate was provided in the manuscript text as well as in Supplemental Fig S1. All these information were given in the Notes of the Taxonomy section of the modified manuscript and reproduced below.
- Notes: The new species, torokii is both morphologically and phylogenetically unique from other described members of the genus Parengyodontium (Tsang et al., 2016;Teixeira et al., 2020). Phenotypically, P. torokii can be readily distinguished from P. album based on conidial shape. P. torokii produces subcylindrical to ellipsoidal conidia, whereas P. album conidia are globose, smooth, hyaline, oval, and apiculate (Tsang et al., 2016). P. torokii differs from P. americanum as the former produces terminal fertile zigzag shaped rachides, but the latter lacks them and produces conidia on right-angled phialides or aphanophialides. In addition, the conidia of P. torokii differ from those of P. americanum in that they are sub cylindrical to ellipsoidal vs. cylindrical to globose in P. americanum (Teixeira et al., 2020). Interestingly, strain CBS 368.72 (subclade 3, sensu Tsang (Tsang et al., 2016)) was morphologically similar to P. torokii in conidial shape and zigzag rachides based on SEM (Figure S1). Based on Maximum Likelihood molecular phylogenetic analyses of the ITS region (Figure S2) as well as three loci analysis (Figure S3), P. torokii is a distinct species as it occurs on a unique clade (subclade3 sensu Tsang (Tsang et al., 2016), Figure S2 and S3). Further, in the five loci analysis, P. torokii, P. album, and P. americanum are seen as distinct clades with moderate to significant statistical support.
- For the Taxonomy section, “Note:” should be add for comparison of the morphology and molecular data of a new species and close-related species.
Ans: Added as suggested. See above.
- The phylogenetic species delimitation, what is the genealogical concordance phylogenetic species recognition (GCPSR) criterion used in this study? Please describe and mention. Figure of phylogenetic tree should be replace with the high resolution.
Ans: We used three different criteria for diagnosing Parengyodontium torokii as a novel species: 1) the ITS, 28S nrDNA, and β-tubulin scheme; 2) The ITS, LSU, SSU, RPB1, RPB2, and TEF1 scheme and; 3) The phylogenomic concept using concordance factors in 5,334 single-copy orthologs. The three methods showed that Parengyodontium torokii is monophyletic and is clustered apart from its closely related species P. album or P. americanum. Thus, showing phylogenetic concordance. The individual gene phylogenies also showed a robust phylogenetic discrimination between species but was not showed in the manuscript since concatenated trees (or species trees) are robust enough to diagnose this new taxon.
High resolution figures are given as .tiff files.
- The discussion should re-write in depth to elaborate discussion.
Ans: We modified to the best of our knowledge and wish to know which part of the discussion needs to be elaborated. Taxonomy section discussion was given first, followed by biofilm, then comparative genomic analysis, finally with metabolome work. The discussion section is already long (~125 lines; 1,650 words). However, if particular portion need to be elaborated, we can provide those information.
- Please verify the whole manuscript carefully for a journal format and editorial issues, such as italic and language mistakes.
Ans: Most of the corrections were incorporated. The italicizing the family name is not allowed as per the Integrative Biology 335 — Nomenclature. https://www.life.illinois.edu/ib/335/nomenclature.html
- Some corrections showed in the attached file.
Ans: All suggested corrections were incorporated in the modified manuscript.
Reviewer 4 Report
Genomic characterization of Parengyodontium torokii sp. nov., a biofilm-forming fungus isolated from Mars 2020 assembly facility
The article is good and suitable for JOF. This research is novel and may be of interest to readers. My observations are as follows:
- Line 45 -46. The authors could modify the wording of the sentence: However, while fungal species also produce protective structures (spores, conidia, or cysts) as both part of their life cycle … Spores may be structures of sexual or asexual reproduction, and they are involved in dispersal and survival.
- Line 181 - 182. The sentence could be modified: “Equal quantities (106 cells or conidia) of myceliumand conidial suspension..” I agree with your wording in the case of conidia. In the case of mycelium it can be confusing.
- Line 188-189: Specify the number of days in the phrase “Vials containing Inconel or Teflon were incubated at 25ËšC in an orbital shaker at 25 RPM for at least 21 days prior to harvesting and analysis.”
- Line 431-437: En la Figura 1 puede indicar el diámetro de la placa Petri utilizada en A y B.
- Line 438-441: It is interesting the detection of thin membranous white layers surrounding the conidia and were presumed to be composed of molecules such as extracellular polymeric substances(EPS), which enabled biofilm formation. I assume that due to the size there is no evidence of this membrane on traditional microscopy.
- The acronyms NRRL and DSMZ may need to be specified.
- Line 617: Figure 6. Comparative genomic analysis of Cordycipitaceae fungi should have better resolution. The information cannot be easily read..
- Line 733-735: The next sentence should be a separate sentence “a careful examination of these two particular strains utilizing polyphasic taxonomy by including more isolates of the related genera is needed.” The authors could also mention what these genera would be.
- Line 744-747: The following sentence seems repetitive with the information from the introduction. During spaceflight, microorganisms may have time to form biofilms within vulnerable systems, causing health and corrosion hazards for space missions. The development of nanoengineered materials that prevent and mitigate biofilm formation is significant to current and future NASA missions. Por favor, revise.
- Line 757-759: You can further discuss the following argument: This minimal amount of mycelium between the coupon and the upper canopy of the biofilm could be due to extracellular polymeric substances (EPS) or other secreted compounds attaching the fungal biomass to the substrate.
Author Response
The article is good and suitable for JOF. This research is novel and may be of interest to readers. My observations are as follows:
- Line 45 -46. The authors could modify the wording of the sentence: However, while fungal species also produce protective structures (spores, conidia, or cysts) as both part of their life cycle … Spores may be structures of sexual or asexual reproduction, and they are involved in dispersal and survival.
Ans: The following sentence is given in the context of survival of fungi in extraterrestrial conditions and hence modifications suggested by the reviewer will be out of context. We wish to keep this sentence as is:
- However, while fungal species also produce protective structures (spores, conidia, or cysts) as both part of their life cycle and as a response to environmental stress, few studies have examined their presence on the spacecraft-associated surfaces or their survival under simulated space conditions.
- Line 181 - 182. The sentence could be modified: “Equal quantities (106 cells or conidia) of myceliumand conidial suspension..” I agree with your wording in the case of conidia. In the case of mycelium it can be confusing.
Ans: Further checking the lab notebook, it is confirmed that we used only conidial spores and not mycelia. Hence, we kept only conidia in the text.
- Line 188-189: Specify the number of days in the phrase “Vials containing Inconel or Teflon were incubated at 25ËšC in an orbital shaker at 25 RPM for at least 21 days prior to harvesting and analysis.”
Ans: 21 days mentioned in this sentence.
- Line 431-437: En la Figura 1 puede indicar el diámetro de la placa Petri utilizada en A y B.
Ans: Petridish diameter in Fig 1 A/B are given (9 cm diameter).
- Line 438-441: It is interesting the detection of thin membranous white layers surrounding the conidia and were presumed to be composed of molecules such as extracellular polymeric substances(EPS), which enabled biofilm formation. I assume that due to the size there is no evidence of this membrane on traditional microscopy.
Ans: Reviewer’s observation is correct. SEM resolution is higher as we know compared to simple optical microscopy.
- The acronyms NRRL and DSMZ may need to be specified.
Ans: Added in the manuscript
- Line 617: Figure 6. Comparative genomic analysis of Cordycipitaceae fungi should have better resolution. The information cannot be easily read.
Ans: The old Figure 6 is split into two figures to increase resolution.
- Line 733-735: The next sentence should be a separate sentence “a careful examination of these two particular strains utilizing polyphasic taxonomy by including more isolates of the related genera is needed.” The authors could also mention what these genera would be.
Ans: Modified as suggested and reproduced below.
- Since kalimantanense BTCC-F23 and T. wallacei CBS 101237 also clustered within the Parengyodontium lineage, a careful examination of these two particular strains utilizing polyphasic taxonomy by including more isolates of the related genera (Lecanicillium and Torrubiella) is needed.
- Line 744-747: The following sentence seems repetitive with the information from the introduction. During spaceflight, microorganisms may have time to form biofilms within vulnerable systems, causing health and corrosion hazards for space missions. The development of nanoengineered materials that prevent and mitigate biofilm formation is significant to current and future NASA missions. Por favor, revise.
Ans: We modified and it should read as below.
- Although not a major concern for robotic missions, long duration crewed missions may give microorganisms adequate time to form biofilms within vulnerable systems, putting both crew health and spacecraft longevity in jeopardy. The development of nanoengineered materials that prevent and mitigate biofilm formation is significant to current and future NASA missions.
- Line 757-759: You can further discuss the following argument: This minimal amount of mycelium between the coupon and the upper canopy of the biofilm could be due to extracellular polymeric substances (EPS) or other secreted compounds attaching the fungal biomass to the substrate.
Ans: The following sentences were given to discuss the argument.
- The confocal stains used were targeting nucleic acids (ToPro-3) and cellulose (Calcofluor-white), thus any secreted EPS compounds lacking nucleic acid or cellulose would not have appeared in these images. It is unclear if material is present, and if so was it secreted after attachment to the surface, or if the initial colonizing mycelium/fungal conidia had a thick layer of a coating (Figure S4A). If the colonizing mycelium/fungal conidia had such an initial coating, this could impart some protection from contact with the antimicrobial coating. SEM analysis of vegetative cells of torokii FJII-L10-SW-P1 strain (Figure 1E-1F, and Figure S4) show the presence of a thin membranous layer that may be linked to this phenomenon.
Round 2
Reviewer 3 Report
Thank you very much for improving the manuscript. The revise version has some typos error please return to author again. See in the attached file.

Author Response
Line # 416: Deleted
Line # 418: Modified as suggested
Line # 444: Three other strains belong to P. album subclade 3 CBS 368.72, UAMH 9836, and LEC01 isolates were isolated from turbine fuel sample, Dayton, Ohio, USA (LEC01); from fresco, Romania (CBS 368.72); and from a human bronchoscopy specimen, Canada.
Line # 447: Deleted.
Figure 3: High resolution figure was submitted to the editor directly. Also, RPB, TEF were given in capital letters.
Figure 5: High resolution figure was submitted to the editor directly.
Line # 671: Parengyodontium sp. is not italicized.